# High-fidelity Cas9-mediated targeting of *KRAS* driver mutations restrains lung cancer in preclinical models

Juan Carlos Álvarez-Pérez [1,2,3,10], Juan Sanjuán-Hidalgo[1,2,10],
Alberto M. Arenas [1,2,3,10], Ivan Hernández-Navas[4,5],
Maria S. Benitez-Cantos [1,3,6], Alvaro Andrades [1,2,3], Silvia Calabuig-Fariñas[7,8],
Eloisa Jantus-Lewintre [8,9], Luis Paz-Ares[5], Irene Ferrer[4,5] &
Pedro P. Medina [1,2,3] ✉

Missense mutations in the 12th codon of *KRAS* are key drivers of lung cancer, with glycine-to-cysteine (G12C) and glycine-to-aspartic acid (G12D) substitutions being among the most prevalent. These mutations are strongly associated with poor survival outcomes. Given the critical role of KRAS in lung cancer and other cancers, it remains as a major target for the development of new and complementary treatments. We have developed a CRISPR-High Fidelity (HiFi)-Cas9-based therapy strategy that can effectively and specifically target *KRAS^{G12C}* and *KRAS^{G12D}* mutants, avoiding *KRAS^{WT}* off-targeting and affecting KRAS downstream pathways, thereby significantly reducing tumorgenicity. The delivery of HiFiCas9 components via ribonucleoprotein particles (RNPs) and adenovirus (AdV) effectively abrogates cell viability in *KRAS*-mutant Non-Small Cell Lung Cancer (NSCLC) preclinical models, including 2D and 3D cell cultures, cell-derived xenografts (CDX), and patient-derived xenograft organoids (PDXO). Our in vitro studies demonstrate that HiFiCas9-based therapy achieves superior KRAS inhibition compared to Sotorasib and effectively circumvents certain resistance mechanisms associated with Sotorasib treatment. Moreover, in vivo delivery using adenoviral particles significantly suppresses tumor growth in preclinical NSCLC models. Collectively, our findings establish HiFi-Cas9 as an effective therapeutic strategy with promising clinical applications, especially if in vivo delivery methods are further optimized.

Accounting for 18% of cancer-related deaths in 2020, lung cancer is the deadliest malignancy worldwide. Estimates suggest its mortality rate will increase to 63.8% by 2040[1]. Indeed, in the US alone, about 350 people die daily from lung cancer, almost 2.5 times more than the number of people dying from colorectal cancer, the second leading cause of cancer death[2]. Non-Small Cell Lung Cancer (NSCLC) is the most common type of lung cancer, and lung adenocarcinoma (LUAD) is its most prevalent subtype.

Despite recent clinical advances, LUAD remains a major concern in the global health landscape, with a 5-year relative survival rate around 20%[1,2]. Therefore, additional research is needed to improve early diagnosis, prognosis, and treatment options. In cancer biology, it has been extensively reported that tumorigenesis often results from the acquisition of driver mutations in certain genes, which leads to uncontrolled cell division, and eventually to the development of cancer[3].

Kirsten rat sarcoma viral oncogene homolog (KRAS) is the most frequently mutated oncogene in human cancer, with a mutation rate of approximately 32% among LUAD patients and 88% in pancreatic adenocarcinomas[4,5]. KRAS is a small GTPase that, under normal conditions, switches between an active state (GTP-bound) and an inactive state (GDPbound) in response to upstream growth factor signaling, thus regulating cell growth, proliferation, and survival[6]. However, single-base missense mutations in Gly12 severely impact its GTP

hydrolysis ability[7–9]. The prevailing tenet is that the hyperactivation of KRAS effectors, including the mitogen-activated protein kinase (MAPK) and phosphatidylinositol 3-kinase (PI3K) pathways, is linked to tumorigenesis, aggressive disease, and a poor prognosis[10–12]. Mouse models of KRAS-induced oncogenesis revealed that tumors regress dramatically once this initial KRAS-oncogenic stimulus ceases[13], being a good example of what was coined as "oncogene addiction". This term describes a state in which cancers remain addicted to, or dependent

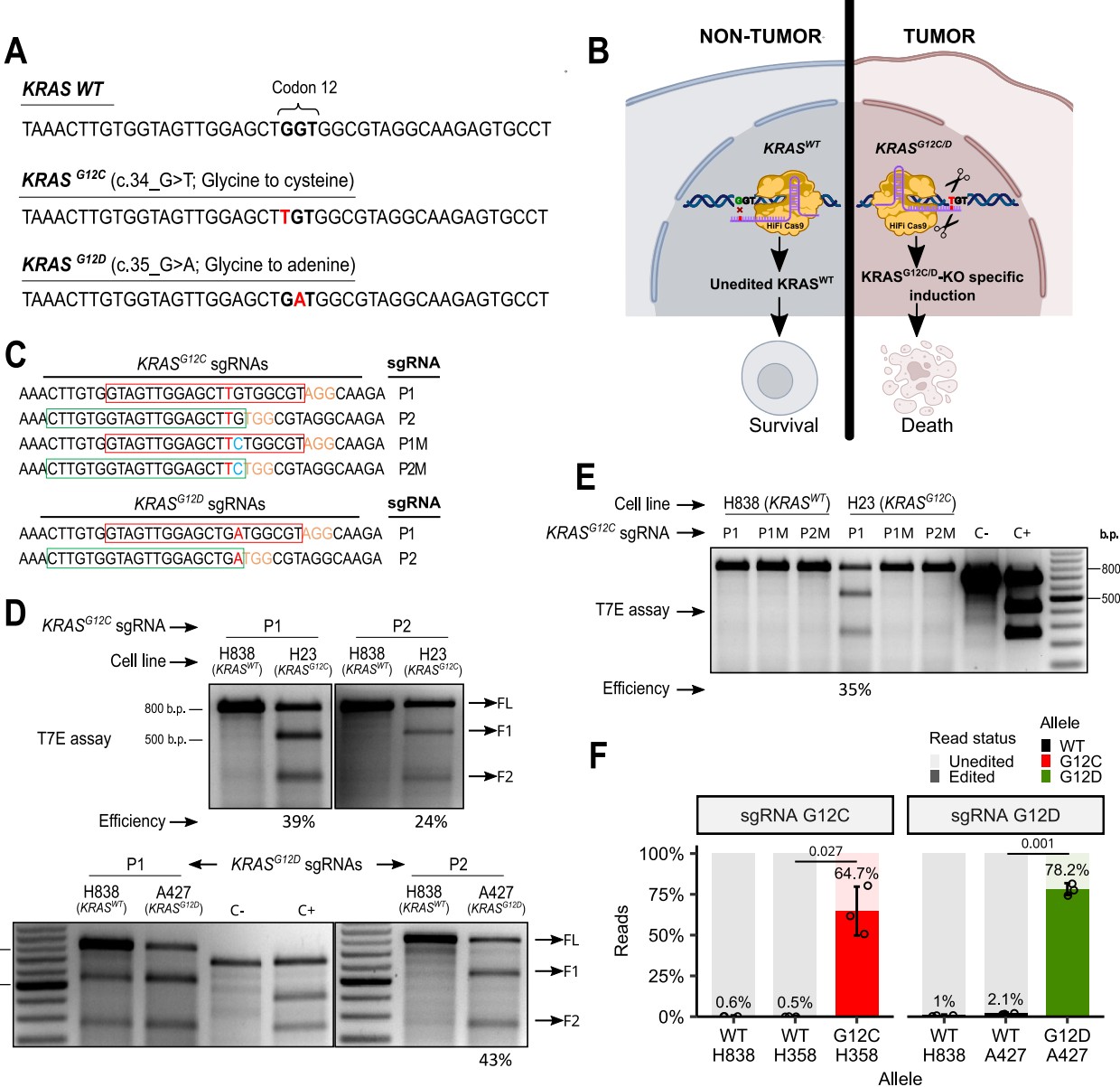

Fig. 1 | Efficiency and specificity of KRAS[mut]-specific sgRNAs. A Representation of KRAS[WT] and KRAS[mut] alleles at genomic DNA level, with the mutated nucleotide highlighted in red. B Graphical representation of the therapeutic approach: KRAS oncogene-addicted cells die off when the KRAS[mut]-specific KO is induced. No effect on KRAS[WT]/non-tumor cells. C sgRNA designs utilizing different protospacer adjacent motifs (red frames for PAM1 and green for PAM2) to target KRAS[mut] alleles. Mutant nucleotides are displayed in red, artificially introduced mismatches in blue, and PAM sequences in orange. D T7-endonuclease assay in KRAS[WT] and KRAS[mut] cell lines. Efficiency calculation of KRAS[mut]-specific sgRNAs on digitized agarose gels after T7-endonuclease assay (shown as percentage below the lane with edition) using the formula: Editing efficiency(%) = $100 \times \left(1 - \sqrt{1 - \frac{(F1+F2)}{(F1+F2+FL)}}\right)$. F1 and F2 refer to the relative pixel density of fragments 1 and 2; FL refers to the full-length

undigested amplicon. Gel images are representative of n = 3. E T7-endonuclease assay in KRAS[WT] and KRAS[G12C] cell lines using sgRNAs with single mismatches. C-: negative control (100% complementary dsDNA). C + : positive control (heteroduplex with indels). The image shown is representative of three independent experiments. F KRAS allele edition frequency by sgRNA. Proportion of unedited and edited targeted high-throughput sequencing reads from either WT, G12C and G12D alleles in heterozygous (H358, A427) and WT homozygous cell lines (H838). Bar graphs represent mean edited read percentages ± SD from three biological replicates for each cell line. Statistical analysis was performed using two-tailed unpaired t-tests with FDR-corrected p values. Source data for panels D, E, and F are provided in the Source Data file.

on, initiating oncogenes and the viability of tumors is dramatically compromised if these oncogenic stimuli are lost[14]. Such oncogene addiction potentially yields important therapeutic opportunities. However, although the importance of *KRAS* mutations in human malignancies has been well-established, to date, no effective anti-cancer therapies specifically targeting *KRAS* mutations have reached the clinic, except for the recently developed KRAS[G12C] inhibitors[15–17]. Because *KRAS* mutations underpin both the initiation and progression of cancer, an ideal therapeutic approach would specifically target these mutations without affecting normal tissue. In this regard, CRISPR-Cas9 technology stands out as a convenient and efficient way to knock out any desired gene. CRISPR systems work by using a single guide RNA (sgRNA) to direct the Cas9 nuclease to make double-stranded breaks (DSBs) in the target DNA. These breaks trigger an error-prone repair mechanism called non-homologous end-joining (NHEJ)[18], which results in random insertions or deletions (indels) that cause frameshifts and premature stop codons, potentially rendering the target gene non-functional.

By utilizing the ease of the CRISPR-Cas9 system, we developed a mutant *KRAS*-specific excision system with therapeutic potential in preclinical models. In this study, we focused on the most prevalent *KRAS* mutations in lung cancer patients, G12C in smokers and G12D in non-smokers[7]. To achieve specificity in our double-targeted therapy, we utilized a high-fidelity version of SpCas9 (HiFiCas9) that maintains high on-target activity while minimizing off-target effects[19]. Although various CRISPR-Cas systems have been employed in previous mutant *KRAS* therapies[20], this study presents, a precise and efficient CRISPR-Cas9 system that targets the oncogenic *KRAS*[G12C] and *KRAS*[G12D] alleles in NSCLC without impacting the wildtype *KRAS*. This study also represents a *KRAS*-editing strategy demonstrating therapeutic potential on preclinical NSCLC models.

## Results

### CRISPR-HiFiCas9 enables robust and highly specific targeting of *KRAS* driver mutations

The *KRAS*[G12C] and *KRAS*[G12D] point mutations differ from the *KRAS*[WT] sequence by only a single nucleotide (Fig. 1A). To ensure the therapeutic approach is selective for cancer cells and minimizes unintended effects in normal tissue, specificity is crucial to avoid off-target interactions with *KRAS*[WT] (Fig. 1B). To achieve this, we systematically evaluated various endonucleases and sgRNAs, aiming to identify a system capable of discriminating between single mismatches, especially those distinguishing *KRAS*[G12C] and *KRAS*[G12D] mutations from *KRAS*[WT].

To evaluate both editing efficiency and specificity, we designed different sgRNAs targeting *KRAS* using two protospacer adjacent motif (PAM) sites: PAM1 (*P1: AGG*) and PAM2 (*P2: TGG*). These sgRNAs were complexed with HiFiCas9 to form ribonucleoproteins (RNPs) (Fig. 1C) and subsequently delivered into *KRAS*-mutant (H23, H358, and A427) and *KRAS*[WT] (H838) cell lines via lipofection. Although the *KRAS*[WT] allele represents the primary off-target concern, neither P2-sgRNA-G12D nor any sgRNAs targeting *KRAS*[G12C] induced detectable editing in *KRAS*[WT] cells. Notably, P1-sgRNA-G12C exhibited the highest editing efficiency (Fig. 1D). However, P1-sgRNA-G12D lacked specificity, leading to unintended editing of *KRAS*[WT] cells (Fig. 1D). Based on their superior editing efficiency and specificity, P1-sgRNA-G12C and P2-sgRNA-G12D were selected for subsequent experiments.

To thoroughly ascertain the extent of specificity, we deliberately introduced a single mismatch adjacent to the mutated nucleotide in the *KRAS*[G12C]-mutant-specific sgRNA, highlighted in blue in Fig. 1C. Subsequent T7-endonuclease assays demonstrated editing only when the appropriate mutant-specific sgRNA was delivered into the cell line carrying the corresponding mutation (Fig. 1E). To ensure an accurate evaluation of specificity, we considered transfection efficiency, utilizing fluorescently labeled tracrRNA (CRISPR-Cas9

tracrRNA-ATTO 550) as a component of these sgRNAs (Supplementary Fig. 1A).

Significantly, none of the non-HiFiCas9-based systems tested were able to discriminate *KRAS* point mutations, regardless of the PAM motif or sgRNA sequence used (Supplementary Fig. 1B). These findings are consistent with the observations made by Kim et al. in 2018[21] and are indicative of the greater specificity of HiFiCas9.

To rule out that the observed specificity was genomic context-dependent, we conducted an additional T7-endonuclease assay on an isogenic panel of *KRAS*-humanized mouse embryonic fibroblast (MEF) models, where the only variable was the *KRAS* variant (*KRAS*[WT], *KRAS*[G12C] or *KRAS*[G12D]) (Supplementary Fig. 2A). Neither of the sgRNAs-*KRAS*[mut] caused DNA editing in *KRAS*[WT] MEFs, and editing was only observed when the corresponding sgRNA-*KRAS*[mut] was delivered into the MEFs harboring the matching *KRAS* mutation (Supplementary Fig. 2C). Similar to the approach used for human cell lines, we also considered transfection efficacy (Supplementary Fig. 2B) These results were further validated through analysis of Sanger sequenced samples using the ICE (Inference of CRISPR Edits) tool (Synthego Performance Analysis, ICE Analysis. 2019. v3.0. Synthego; https://ice.synthego.com/#/). ICE results confirmed the high accuracy of our system observed with T7 endonuclease assays (Supplementary Fig. 2D).

Next, we performed next-generation sequencing (NGS) on PCR-amplified *KRAS* amplicons from RNP-lipofected samples to characterize the nature of the indels generated and confirm editing efficiency and specificity. Our targeted NGS analysis revealed key findings: (i) *KRAS*[G12C]-sgRNA and *KRAS*[G12D]-sgRNA specifically targeted the *KRAS*[mut] alleles while leaving the *KRAS*[WT] alleles unaffected (Fig. 1F); and (ii) indel distribution was highly consistent across replicates and comparable between *KRAS*[G12C] and *KRAS*[G12D] mutant cell lines (Supplementary Fig. 3A).

Deletions were the most frequent outcome of *KRAS* gene editing. Among the deletion events in H358 (*KRAS*[G12C]), the loss of 2 nucleotides was the most prevalent. In A427 (*KRAS*[G12D]) cells, although deletions were also the main outcome, the most representative event was the insertion of 1 nucleotide. Hardly any indel events were detected in H838 (*KRAS*[WT]) cells (Supplementary Fig. 3A).

These findings demonstrate that the use of mutation-specific sgRNAs leads to the disruption of the ORF (Open Reading Frame) of mutant *KRAS* without altering its wild-type version.

To investigate potential off-target effects outside of the *KRAS* gene region, we used the Cas-OFFinder[22] and Off-Spotter[23] algorithms. As expected, both algorithms predicted *KRAS*[WT] allele as the most probable off-target (Supplementary Fig. 4A−B). Additionally, genes with up to three mismatches in the 10 nucleotides upstream of the PAM sequence were considered potential off-targets[24]. Using that criterion, the algorithms were able to predict *ABI3BP* and *SIPA1L3* as potential off-targets with three mismatches for sgRNA-*KRAS*[G12D], while no coding off-targets were reported for sgRNA-*KRAS*[G12C] (Supplementary Fig. 4A). To assess off-targeting activity in these loci, we infected H838 (*KRAS*[WT]) and H358 (*KRAS*[G12C]) cells with CRISPR-containing AdVs (generated for in vivo experiments and explained in the next section) and used the ICE and TIDE (Tracking of Indels by Decomposition)[25] tools to analyze the results. Our analysis showed no evidence of unintended damage to *ABI3BP, SIPA1L3*, or *KRAS*[WT], indicating that the designed sgRNAs specifically targeted the *KRAS* mutations (Supplementary Fig. 4C).

### HiFiCas9-mediated *KRAS* targeting impairs the viability of mutant cell lines

The validated targeting systems were utilized to evaluate the functional impact of disrupting mutant *KRAS* in NSCLC cells. First, RNP complexes were introduced through lipofection into a panel of heterozygous *KRAS*-mutant NSCLC cell lines including A427 and SK-LU-1 (*KRAS*[G12D]), H1792 and H358 (*KRAS*[G12C]), and H838 (*KRAS*[WT]). Cell

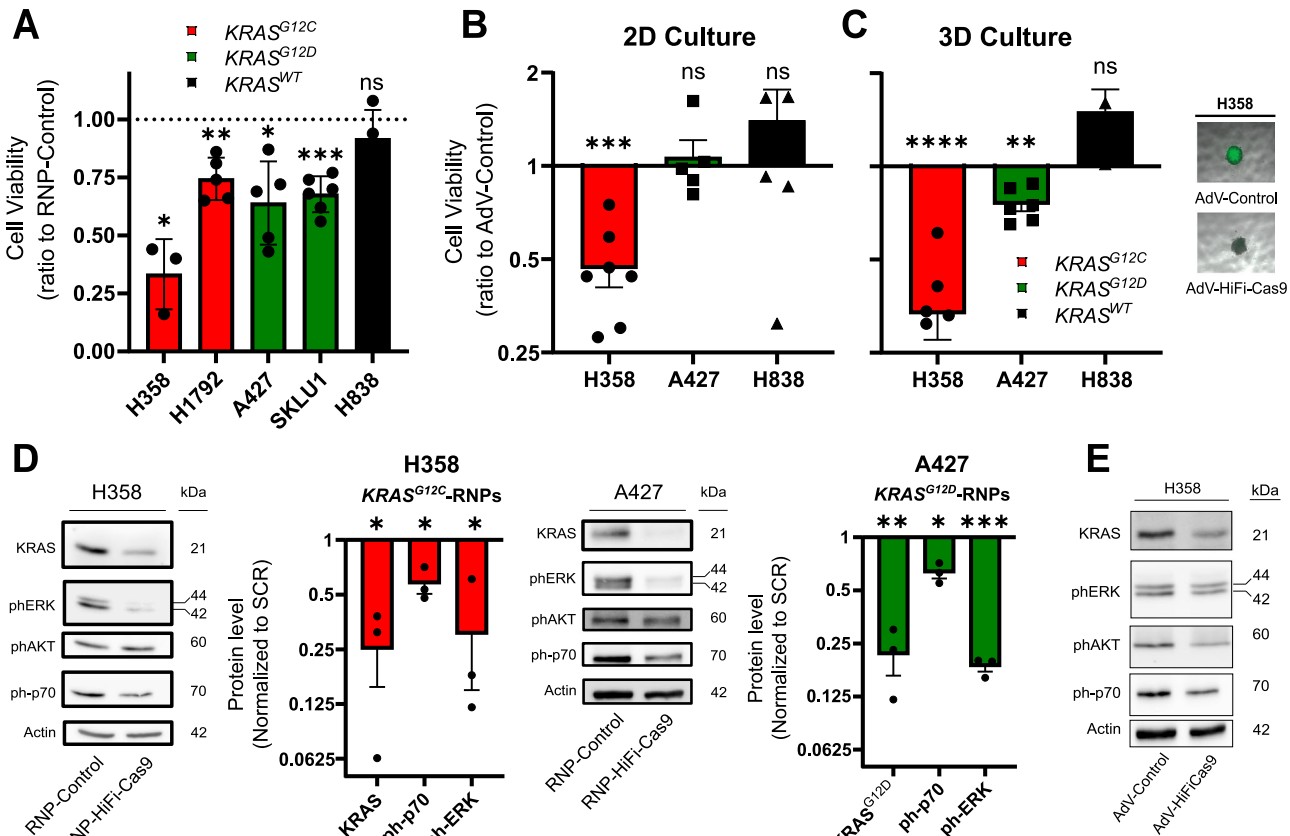

**Fig. 2 | Genome editing of *KRAS^G12C* and *KRAS^G12D* impairs viability of tumor cells in vitro. A** Cell viability of *KRAS^mut* NSCLC cell lines targeted with mutation-specific RNPs, 7 days after transfection. Data represent the mean ± SD of *N* independent biological replicates per cell line, analyzed using a two-tailed one-sample t-test (H358, *n* = 3, *p* = 0.016; H1792, *n* = 5, *p* = 0.0034; A427, *n* = 5, *p* = 0.011; SKLU1, *n* = 6, *p* = 0.0002; H838, *n* = 4, *p* = 0.27). Source data are provided as a Source Data file. **B** 2D cell viability assay of NSCLC cell lines 10 days after AdV transduction. Data represent the mean ± SEM of *N* independent biological replicates per cell line, analyzed using a two-tailed one-sample t-test (H358, *n* = 7, *p* = 0.0001; A427, *n* = 5, *p* = 0.65; H838, *n* = 7, *p* = 0.31). **C** Left: 3D cell viability assay of NSCLC cell lines 10 days after AdV transduction. Right: representative 3D spheroid culture of H358 cells. Data represent the mean ± SEM of *N* independent biological replicates per cell

line, analyzed using a two-tailed one-sample t-test (H358, *n* = 7, *p* = 0.0001; A427, *n* = 6, *p* = 0.0014; H838, *n* = 6, *p* = 0.12). **D** Western Blots displaying KRAS-dependent signaling 72 h after RNP transfection. Representative Western blot images (left) and corresponding densitometric quantification (right). Data represent mean ± SEM from three independent biological replicates, analyzed using a two-tailed one-sample t-test (H358: KRAS *p* = 0.016, ph-p70 *p* = 0.025, ph-ERK *p* = 0.045; A427: KRAS^G12D *p* = 0.0044, ph-p70 *p* = 0.016, ph-ERK *p* = 0.0003). **E** Western Blot displaying KRAS-dependent signaling in H358 cells 72 h after AdV transduction. (Representative image of two biological replicates). Treatment was administered once at time point t = 0 in all the experiments. Source data for panels A-E are provided in the Source Data file.

viability was then assessed using the CellTiterGlo kit following the manufacturer's instructions (See Materials and Methods section). The results showed a decrease in cell viability for *KRAS^G12C* cells (67% for H358 and 26% for H1792) and *KRAS^G12D* cells (36% for A427 and 32% for SK-LU-1) 7 days after RNP transfection (Fig. 2A). No significant reduction in cell proliferation was observed in *KRAS^WT* H838 cells (Fig. 2A).

To translate the therapeutic strategy to an in vivo xenograft setting, we generated AdVs containing the validated CRISPR components, including a multiplexed double sgRNA that targets both *KRAS* mutations and HiFiCas9 (Supplementary Fig. 5A). Prior to assaying the AdV-based system in vivo, we tested it in 2D and 3D cell cultures. Usually, 2D cell cultures fail to replicate key aspects observed in vivo tumors[26], which can be better replicated by 3D cultures[27]. Therefore, we compared monolayer (2D-adherent) and 3D ultra-low adherent spheroid cultures in our in vitro AdV system. The efficacy of the AdV system was evaluated in H358 and A427 cell lines through the determination of the infection efficiency by GFP fluorescence and the editing efficiency by Sanger sequencing (Supplementary Fig. 5B–C). In alignment with previous studies on KRAS^G12C inhibitors[17,28], the system showed greater efficacy in 3D spheroids, more accurately mimicking in vivo tumors.

The results showed a reduction in cell viability from approximately 50% in 2D to nearly 70% in 3D for H358 and from nearly unchanged in 2D to nearly 25% in 3D for A427, while the viability of H838 (*KRAS^WT*) cells remained unchanged in both culture systems (Fig. 2B−C). Remarkably, these results were achieved despite relatively modest levels of transduction (~46.6% and 30% for H358 and A427, respectively, Supplementary Fig. 5B).

Next, we examined the expression of *KRAS* and its downstream signaling pathways by Western blot. Treatment with HiFiCas9 RNPs for 72 h led to a substantial decrease in the expression of total KRAS protein in H358 due to the targeting of *KRAS^G12C*, while a complete knock-out of *KRAS^G12D* was observed in A427, as confirmed using a *KRAS^G12D* specific antibody (Fig. 2D). We also noted reduced phosphorylation of KRAS-signaling proteins such as p70S6K and ERK in both cell models, although we did not observe a consistent change in AKT phosphorylation as reported in previous studies[28,29] (Fig. 2D). Results from the AdV-treated cells showed a milder reduction (Fig. 2E), which is likely due to the lower efficiency of the system compared to RNPs. These findings provide evidence that selective targeting of mutant *KRAS* impacts cell growth by suppressing KRAS downstream pathways that regulate cell proliferation.

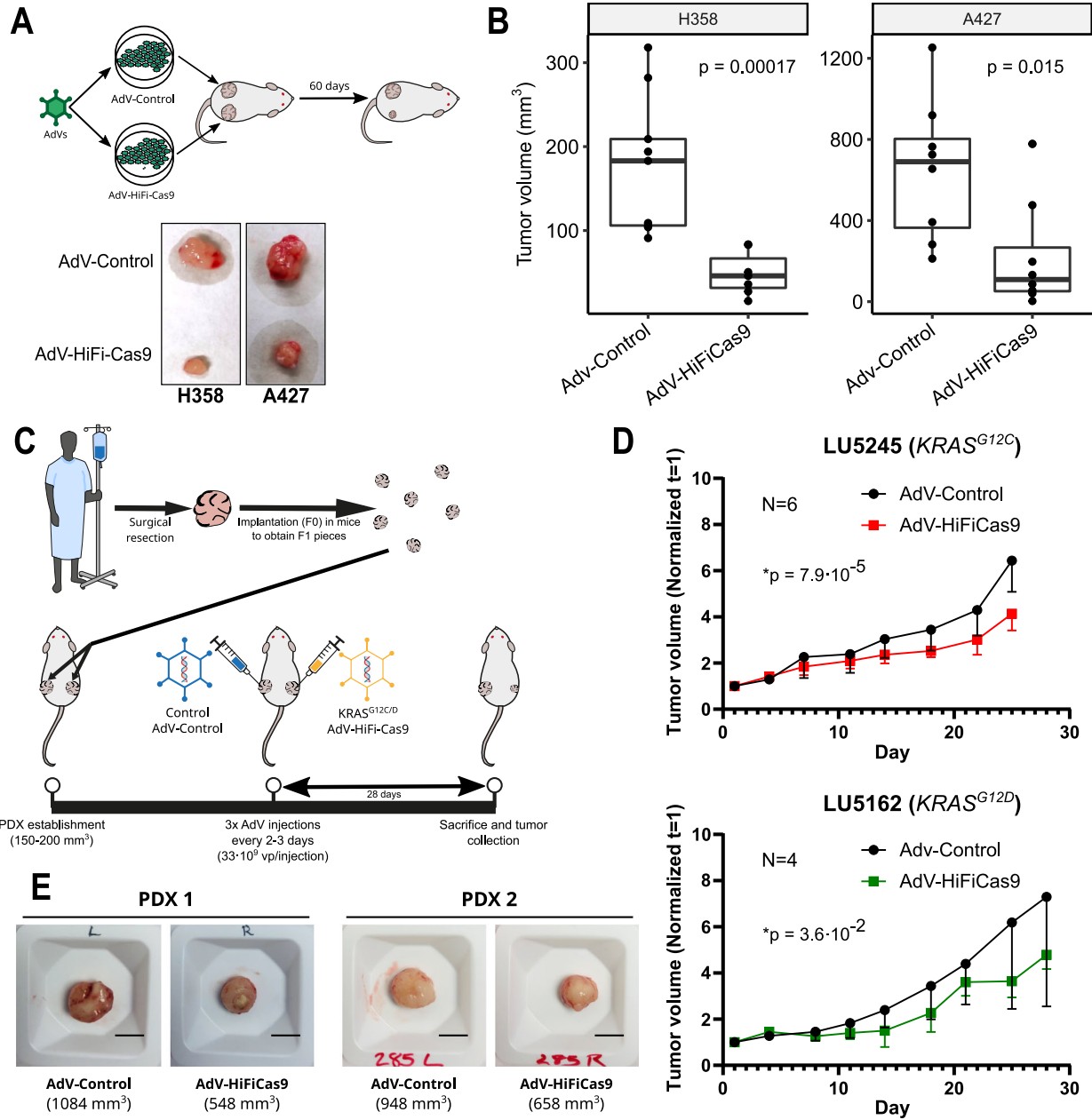

**Fig. 3 | Edition of mutant KRAS induces tumor growth inhibition in PDX and CDXs models. A** Schematic representation of experimental design with representative ex vivo images of tumors harvested 60 days after implantation. $10^6$ pretreated cells were transplanted subcutaneously and tumor growth was monitored for two months. Representative images of $n = 9$. **B** Quantification of ex vivo tumor volumes of H358 (Adv-Control, $n = 9$; Adv-HiFiCas9, $n = 7$) and A427 (Adv-Control, $n = 8$; Adv-HiFiCas9, $n = 8$) CDXs. 63% reduction in tumor volume for H358 (left; p-value: 0.00017) and a - 42% reduction for A427 (right; p-value: 0.015). Boxplots display the median (50th percentile, center line), the 25th and 75th percentiles (lower and upper box bounds, respectively), and the whiskers extend to the smallest and largest values within 1.5 times the interquartile range (IQR) from the lower and upper quartiles. **C** Schematic representation of PDX experimental design. **D** Tumor volumes of LU5245 (G12C; Adv-Control, $n = 6$; Adv-HiFiCas9, $n = 6$) and LU5162 (G12D, Adv-Control, $n = 4$; Adv-HiFiCas9, $n = 4$) PDXs. Mean tumor volume (mm³) normalized to day 1, accompanied by standard deviation. Tumor growth was modeled using a linear mixed-effects model with random intercepts. **E** Ex vivo pictures from $KRAS^{G12C}$ tumors extracted 28 days post-treatment. Scale bar=10 mm. Diameters (mm): PDX1: Control=14, HiFiCas9 = 11; PDX2: Control=13.6, HiFiCas9 = 11.8. Volumes (mm³): PDX1: Control=1084, HiFi-Cas9 = 548; PDX2: Control=948, HiFi-Cas9 = 658. Source data for panels B and D are provided in the Source Data file.

## HiFiCas9-mediated *KRAS* targeting suppresses tumor growth in lung cancer xenograft models

To assess the efficacy of *KRAS* HiFiCas9-based therapy in vivo, we first turned to a xenograft model. In order to minimize experimental variability, we ensured that both xenografts were treated at the same size and that the transduction efficiency was comparable between cell lines and viruses. To achieve this, we transduced $1 \times 10^6$ cells (MOI = 1000 vp/cell) in vitro 2 h prior to transplantation (Fig. 3A).

Subsequently, we subcutaneously transplanted H358 ($KRAS^{G12C}$) and A427 ($KRAS^{G12D}$) cells that had been pretreated with either AdV-Control or AdV-HiFiCas9, with the former on the left flank and the latter on the right flank (Fig. 3A).

We monitored tumor growth over two months and then euthanized the mice to calculate tumor volumes ex vivo (See Materials and Methods). Similar to the in vitro results, we obtained significant tumor growth inhibition in both H358 and A427 models (Fig. 3B). Specifically,

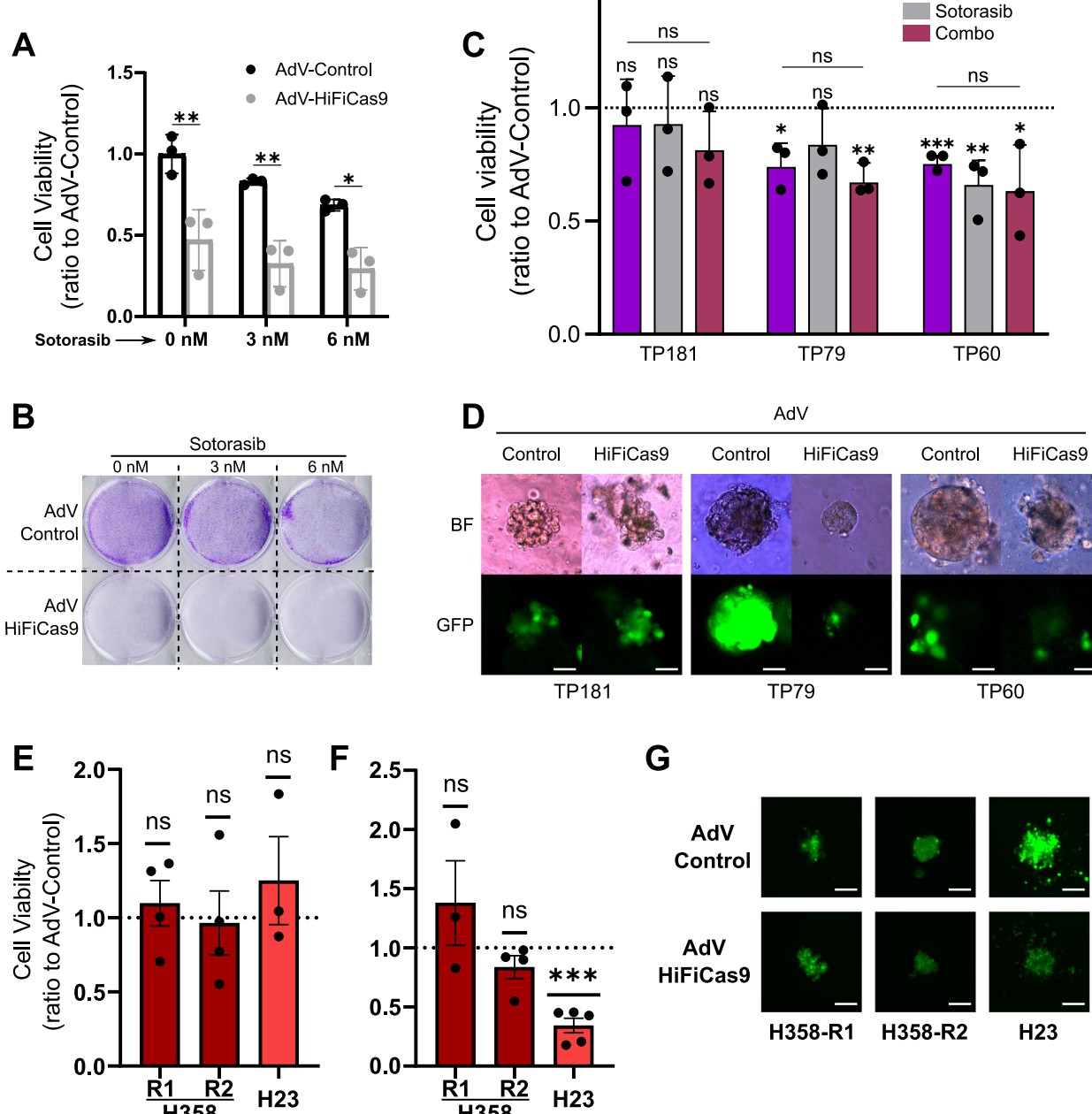

**Fig. 4 | Comparison of Sotorasib and HiFiCas9 therapy. A** Cell viability assay combining HiFiCas9 therapy and Sotorasib in parental H358 cells ($n = 3$ independent biological replicates). Control/HiFiCas9 infection was performed 24 h before Sotorasib treatment. Data represent the mean ± SD of three independent biological replicates, analyzed using a two-way ANOVA. (AdV-Control vs AdV-HiFiCas9; 0 nM Sotorasib $p = 0.0005$, 3 nM Sotorasib $p = 0.0008$, 6 nM Sotorasib $p = 0.0056$) **B** Colony formation assay combining HiFiCas9 therapy and Sotorasib in parental H358 cells. **C** Cell viability of PDXOs treated with HiFiCas9 therapy and/or Sotorasib. Data represent the mean ± SD of three independent biological replicates derived from three different PDXs, analyzed using a two-tailed unpaired t-test. TP79: AdV-HiFiCas9, $p = 0,0133$, Combo, $p = 0.003$; TP60: AdV-HiFiCas9,

$p = 0.0004$, Sotorasib, $p = 0.0058$, Combo, $p = 0.04$. **D** Representative images of organoid cultures five days post-treatment with adenovirus control or HiFiCas9. Scale bar = 100 μm. **E** Cell viability of Sotorasib-resistant cells treated with HiFiCas9 therapy in 2D cultures. Data represent mean ± SEM from $N$ independent biological replicates, analyzed using a two-tailed one-sample t-test (H358R1-R2, $n = 4$; H23, $n = 3$). **F** Cell viability of Sotorasib-resistant cells treated with HiFiCas9 therapy in 3D cultures. Data represent mean ± SEM from $N$ independent biological replicates, analyzed using a two-tailed one-sample t-test (H358-R1, $n = 3$; H358-R2, $n = 4$; H23, $n = 5$, $p = 0.0005$). **G** Representative images of 3D cultures of cells ten days post-treatment with Adenovirus control or HiFiCas9. Scale bar = 50 μm. Source data for panels **A, C, E** and **F** are provided in the Source Data file.

we found a substantial ~63% reduction in tumor volume for H358 tumors (Fig. 3B (left); $p = 0.00017$) and a ~42% reduction in A427 models (Fig. 3B (right); $p = 0.015$).

To further evaluate the molecular effects of *KRAS* editing, we analyzed the MAPK pathway by measuring phosphorylated MEK and ERK levels in $KRAS^{G12C/D}$-derived xenografts (Supplementary Fig. 7A).

However, no clear or consistent patterns were observed, likely due to tumor-specific variations in proliferation. These results suggest that by 60 days post treatment, differences in proliferative signaling pathways may no longer be detectable, possibly due to the persistence of an unedited tumor subpopulation contributing to residual tumor growth.

## HiFiCas9-mediated *KRAS* targeting inhibits tumor growth in Patient-Derived Xenografts (PDXs)

To further evaluate the translatability of our therapeutic strategy, we utilized a Patient-Derived Xenograft (PDX) preclinical model. In PDX models, patient tumors are implanted into immunodeficient mice, allowing for a more clinically relevant assessment of treatment efficacy[30]. For this study, we used two NSCLC PDX models, each harboring a different *KRAS* mutation: LU5162 (*KRAS*^G12D) and LU5245 (*KRAS*^G12C).

Briefly, 2×2 mm tumor fragments were implanted into mice, and once tumors reached an average volume of 150-200 mm$^3$, animals were randomized into treatment groups. Each mouse received 1×10$^{11}$ viral particles of AdV-HiFiCas9 or AdV-Control, administered through three intratumoral injections spaced 2–3 days apart (Fig. 3C). Tumor volumes were measured twice weekly for 28 days (see Methods).

To analyze tumor growth, we used linear mixed-effects models with random intercepts to model the log-tumor volume over time per tumor (see Methods). Our analysis found a significant reduction in tumor growth rate for AdV-HiFiCas9-treated PDXs compared to AdV-Control PDXs in both *KRAS*^G12D ($p = 3.6 \cdot 10^{-2}$) and *KRAS*^G12C ($p = 7.9 \cdot 10^{-5}$) modes (Fig. 3D–E). Taken together, these results demonstrate that in vivo CRISPR-based *KRAS* editing significantly suppresses tumor growth in NSCLC-derived PDX models.

To provide additional context, Supplementary Fig. 6A presents individual tumor volume measurements (mm$^3$) over time without normalization, illustrating the variability observed at the experiment's onset. Supplementary Fig. 6B includes all available *KRAS*^G12C tumor photographs.

Further characterization of treated PDX tumors was performed through Ki67 and Cleaved Caspase 3 immunohistochemistry. As shown in Supplementary Fig. 7B, no significant differences in proliferation (Ki67) or apoptosis (Cleaved Caspase 3) markers were detected. This suggests that tumor reduction was driven by the elimination of edited cells, while the residual tumor mass at 28 days post-treatment consisted largely of non-transduced, unedited cells.

## HiFiCas9-mediated *KRAS* targeting better addresses the oncogenic inhibition of KRAS than the KRAS^G12C inhibitor Sotorasib

Our HiFiCas9-based therapeutic strategy is designed to target two *KRAS* mutations, *KRAS*^G12C, which is targeted by an FDA (Food and Drug Administration) -approved drug, and *KRAS*^G12D, which currently lacks any approved targeted treatment options. In the last years, covalent inhibitors targeting KRAS^G12C have been developed and entered human trials (NCT03600883, NCT03785249). Hence, we aimed to compare the efficacy of our HiFiCas9-based strategy with Sotorasib, the first KRAS^G12C inhibitor approved by the FDA and the European Commission, whose clinical benefits were reported recently in the CodeBreaK100 phase 2 clinical trial (NCT03600883)[31].

We first verified that the drug was effective by confirming the reported IC$_{50}$ for H358 cells (6 nM)[28] (Supplementary Fig. 8A). Next, we infected H358 cells with either Control-GFP or HiFiCas9 loaded AdV particles 24 h prior to the addition of Sotorasib. The growth inhibition observed with HiFiCas9 was greater than that induced by 6 nM (IC$_{50}$) or 3 nM (1/2 IC$_{50}$) of Sotorasib in Control-GFP-treated cells (Fig. 4A, B).

A two-way ANOVA was conducted to assess the effect of Sotorasib and HiFiCas9 treatment on cell viability. No statistically significant interaction between these two independent variables was observed [$F_{(2, 12)} = 0.55$, $p = 0.5899$]. Individually, HiFiCas9 treatment had a highly significant impact on viability ($F_{(1, 12)} = 69.55$, $p < 0.0001$), accounting for 72.96% of the total variance. Sotorasib, although explaining only 13.3% of the variance, also showed statistical significance ($F_{(2, 12)} = 6.34$, $p = 0.0132$). A Tukey post-hoc test revealed significant differences between Control/HiFiCas9 treatment across all Sotorasib concentrations. Interestingly, none of the Sotorasib concentrations significantly further reduced the viability of HiFiCas9-treated cells (Fig. 4A; 6 nM, $p = 0,5062$). Conversely, neither of the used concentrations of Sotorasib precluded a significant therapeutic improvement when combined with the HiFiCas9 treatment.

These results may be explained by the distinct mechanisms through which HiFiCas9 and Sotorasib target KRAS^G12C. HiFiCas9 prevents KRAS^G12C synthesis at the genetic level, whereas Sotorasib inhibits the mutant protein only after it has been synthesized and is bound to GDP. This fundamental difference in their modes of action likely accounts for the lack of synergy observed when combining Sotorasib with HiFiCas9, as the absence of KRAS^G12C protein leaves no substrate for Sotorasib to act upon. This lack of synergy was further confirmed in a clonogenic assay, which demonstrated the superior efficacy of HiFiCas9 alone (Fig. 4B).

## Comparative analysis of HiFiCas9-mediated *KRAS*^G12C targeting and Sotorasib treatment in PDX-derived organoids (PDXO)

To assess the therapeutic potential of genomic *KRAS*^G12C elimination in a preclinical setting that more closely resembles the clinic, we compared the efficacy of HiFiCas9-mediated *KRAS*^G12C targeting with Sotorasib treatment in patient-derived xenograft organoid (PDXO) models. The experimental design included four treatment conditions: AdV-Control, AdV-HiFiCas9, AdV-Control + Sotorasib, and AdV-HiFiCas9 + Sotorasib. Cell viability was measured on day 1 (t = 1) and after seven days (t = 7) using the CellTiter-Glo assay, with data normalized to t = 1 and the AdV-Control condition. GFP fluorescence confirmed successful infection of all three PDXO models (TP181, TP79, and TP60, Fig. 4D).

Cell viability analysis revealed a significant ~30% reduction in TP79 and TP60 organoids following AdV-HiFiCas9 treatment compared to AdV-Control organoids (Fig. 4C). Consistent with our cell line data, the combined treatment (AdV-HiFiCas9 + Sotorasib) did not produce an additional reduction in viability beyond Adv-HiFiCas9 alone. Notably, TP60 responded significantly to both Sotorasib and AdV-HiFiCas9, while TP79 responded exclusively to AdV-HiFiCas9, highlighting the added therapeutic advantage of our system over Sotorasib. In contrast, TP181 showed no significant viability changes across treatment condition (Fig. 4C), suggesting that this model may be independent of KRAS^G12C activity. Visual inspection further confirmed a marked decrease in organoid size in TP79 and TP60 following AdV-HiFiCas9 treatment, while TP181 remained unaffected (Fig. 4D).

These three PDXO models may represent distinct clinical scenarios: (i) KRAS^G12C-independent tumors (TP181), (ii) tumors with high KRAS^G12C-GTP activity that are not sensitive to Sotorasib[32] but sensitive to *KRAS*^G12C DNA elimination (TP79), and (iii) highly KRAS^G12C-dependent tumors (TP60). Notably, the latter scenario aligns with findings from the CodeBreaK 100 trial, which reported a 32% response rate to Sotorasib in NSCLC patients[33].

Overall, our findings underscore the therapeutic potential of the *KRAS*^mut-CRISPR/HiFiCas9 system for treating *KRAS*-mutant tumors, particularly those resistant to current *KRAS*^G12C inhibitors.

## HiFiCas9-mediated *KRAS*^G12C targeting efficacy in Sotorasib-resistant lung cancer cells

To evaluate whether our CRISPR-based approach provides a therapeutic advantage over Sotorasib-resistant cells, we tested it on three resistant *KRAS*^G12C NSCLC cell lines: H23 and H358-R1 (developed in Eloisa Jantus Lewintre's research group) and H358-R2 (developed in Irene Ferrer's group).

We treated these resistant cell lines with our adenoviral-based HiFiCas9-mediated *KRAS*^G12C targeting therapy in both 2D-adherent cultures and 3D-spheroid models. While no significant differences in cell viability were observed in 2D cultures (Fig. 4E), the 3D spheroid culture of H23-resistant cells exhibited a statistically significant 60% reduction in viability across six independent experiments ($n = 6$, Fig. 4F). These findings suggest that high levels of GTP-bound KRAS^G12C

may contribute to resistance against Sotorasib, but that our approach can still effectively reduce viability in certain resistant models. Notably, H23 cells harbor co-occurring mutations in *KEAP1* and *SMARCA4*, which have been linked to poor clinical outcomes in Sotorasib-treated patients[34,35].

Additionally, we performed further characterization of the indel distribution (Supplementary Fig. 3B), infection efficiency (Fig. 4G), and *KRAS* targeting in all three resistant cell lines (Supplementary Fig. 3C). These results highlight the potential of our strategy to overcome KRAS[G12C] inhibitor resistance in a subset of NSCLC models.

### Resistance assays of the HiFiCas9-mediated *KRAS* targeting therapy

A major challenge in KRAS[G12C]-targeted therapy is the development of resistance over time. To determine whether our CRISPR-based strategy exhibits signs of resistance or adaptation, we conducted a longitudinal experiment involving multiple rounds of *KRAS* editing. Parental H358 and A427 cells were initially treated with RNPs containing either Control/scrambled-gRNA or *KRAS[G12C/D]*-gRNA, and cell viability was assessed over 7 days. Following recovery and approximately 30 days of regrowth, *KRAS[G12C/D]*-gRNA-treated cells underwent a second round of RNP transfection, with viability measurements repeated. This cycle was performed a third time (Supplementary Fig. 8B).

Across all three consecutive treatments, we observed a consistent loss of viability, with no indication of resistance development (Supplementary Fig. 8C). Sanger sequencing further confirmed that *KRAS* alterations remained evenly distributed across all batches, with no evidence of resistance-associated indels (Supplementary Fig. 8D).

We hypothesize that if resistance or adaptation were to occur, it would likely involve positive selection of indels that preserve oncogenic activity while escaping *KRAS[G12C/D]*-gRNA recognition and re-editing. However, our results indicate that genetic elimination of mutant *KRAS* remains an effective approach and could potentially overcome resistance mechanisms observed in Sotorasib-resistant tumors.

## Discussion

The reliability and efficacy of CRISPR-Cas9 technology for gene knockouts have led to the possibility of leveraging this system to specifically target oncogenic mutations as an antineoplastic therapy. In this study, we took advantage of a high-fidelity version of Cas9 (HiFiCas9)[19] to devise a multiplexed system to target effectively and specifically *KRAS[G12C]* and *KRAS[G12D]* driver mutations found in NSCLC while preserving *KRAS[WT]* from off-targeting. We demonstrated the therapeutic value of this approach in cell lines, PDXO models, and in vivo preclinical models, including CDX and PDX.

Therapies targeting DNA-level mutations have the potential to be effective for a wide range of oncogenic mutations, and this study constitutes a proof of concept for the design of a system that can be programmed to target two different *KRAS* oncogenic mutations effectively and specifically. Indeed, unlike traditional protein-level treatments, DNA-level therapies demand exceptional precision, as off-target effects can lead to unintended and lasting consequences. Previous attempts to target KRAS-dependent tumors using CRISPR systems were restricted in specificity and did not provide adequate evidence of specific *KRAS[mut]* targeting, since they either aimed to target all *KRAS* variants or inadequately evaluated *KRAS[WT]* targeting[21,36,37]. In contrast, our study thoroughly evaluated the specificity of our HiFiCas9 system by (i) interrogating the specificity of HiFiCas9 with the introduction of mismatches next to the mutated nucleotide (Fig. 1E), (ii) examining the editing of *KRAS* in *KRAS[WT]* cell lines and the *KRAS[WT]* allele in heterozygous cells (Fig. 1D–F and Supplementary Fig. 3), (iii) validating the specificity of both *KRAS[G12C]* and *KRAS[G12D]*-targeting systems in an isogenic humanized model (Supplementary Fig. 2), and (iv) analyzing predicted off-targets effects outside

the *KRAS* gene region (Supplementary Fig. 4). Although other studies specifically targeted *KRAS[G12S]*[38] and *KRAS[G12V]*[39] mutations in lung and pancreas, respectively, our study demonstrates in vitro and in vivo therapeutic potential in patient-derived models (PDXOs and PDXs).

Our data indicate that disruption of either *KRAS[G12C]* or *KRAS[G12D]* driver mutations inhibits cell growth across all tested cell lines in vitro, with no cell toxicity or editing observed in *KRAS[WT]* cells (Fig. 2A). Consistent with previous reports suggesting that *KRAS[mut]* dependency may be underestimated in 2D cultures, our viability assays in both 2D and 3D models demonstrated increased KRAS dependency in 3D cultures[17] (Fig. 2B–C). The observed decrease in viability correlates with the deactivation of KRAS-dependent pathways, as evidenced by decreased phosphorylation of Erk1/2 and p70S6K following *KRAS[mut]* targeting (Fig. 2D–E).

We further validated the therapeutic potential of our approach in vivo using CDXs and PDXs (Fig. 3), which better recapitulate tumor heterogeneity and the tumor microenvironment, offering a predictive accuracy of clinical responses exceeding 80%[40].

After decades of research, Amgen and Mirati Therapeutics have developed two direct KRAS[G12C] inhibitors, Sotorasib (AMG 510 or Lumakras)[16,28] and Adagrasib (MRTX849 or Krazati)[15,41,42]. Sotorasib was the first FDA-approved KRAS[G12C]-specific inhibitor for the treatment of patients with advanced NSCLC who are *KRAS[G12C]*-positive and have received at least one prior systemic therapy, based on the phase 1/2 CodeBreaK 100 study (NCT03600883). Adagrasib was also approved at the end of 2022 for the treatment of patients previously treated with chemotherapy and anti-PD-1/PD-L1 therapy, based on KRYSTAL-1 (NCT03785249). However, their clinical benefit as second-line treatment appears to be lower (objective response rate of 36% to 43%)[16,31,41,43] than those expected from targeted therapies (60% to 70%)[44,45]. Moreover, in comparison with docetaxel, another second-line therapy, Sotorasib did not demonstrate an improvement in overall survival[46], which led to the rejection of Sotorasib treatment as a supplemental new drug application by the FDA in 2024. As of January 2025, the FDA has again approved Sotorasib in combination with panitumumab for *KRAS[G12C]* metastatic colorectal cancer based on the phase 3 CodeBreaK 300 trial (NCT05198934). Currently, these drugs do not work as well as chemoimmunotherapy and so far, are not ready to become first-line agents[47].

The effectiveness of KRAS[G12C] inhibitors is inherently limited by their mechanism of action. These drugs covalently bind to the mutant cysteine residue, locking KRAS[G12C] in its inactive GDP-bound state. However, mutant KRAS predominantly exists in its active, GTP-bound form, which drives oncogenic signaling. Consequently, KRAS[G12C] inhibitors require GTPase activity to function effectively[15,48,49]. This dependency creates vulnerabilities, as resistance can arise through the accumulation of active KRAS and the reactivation of ERK signaling due to the compensatory activation of receptor tyrosine kinases (RTKs) and SOS1/2[50,51]. Resistance mechanisms also include acquired secondary *KRAS* mutations and high-level amplification of the *KRAS[G12C]* allele[17,52,53]. Many of these adaptive and resistance pathways ultimately depend on the relentless expression of *KRAS[G12C]*. The newly synthesized mutant protein can undergo rapid nucleotide exchange in response to suppressed ERK signaling[54,55] or acquire new mutations that (i) block its GTPase activity[53], (ii) confer resistance to these inhibitors[52,56], or (iii) enhance intrinsic and guanine nucleotide exchange factor (GEF)–mediated nucleotide exchange, weakening the potency of KRAS[G12C] inhibitor[57]. These limitations underscore the need for strategies that block KRAS[G12C] production entirely. For this reason, we hypothesized that HiFiCas9-mediated KRAS[G12C] targeting could outperform Sotorasib.

Indeed, HiFiCas9-mediated KRAS[G12C] targeting demonstrated full therapeutic efficacy in both H358 cells and some PDXOs, whereas Sotorasib provided only a marginal, non-significant additional benefit (Fig. 4A–D). This suggests that HiFiCas9 effectively depletes the

oncogenic driver, rendering Sotorasib's inhibition of residual KRAS[G12C] activity negligible. Notably, PDXO data highlighted the therapeutic advantage of our approach in tumors capable of restoring high KRAS-GTP activity despite Sotorasib treatment (e.g., TP79). This aligns with findings in Sotorasib-resistant H23 cells, where HiFiCas9 treatment reduced viability, indicating that high GTP-bound KRAS[G12C] levels may contribute to resistance but can be effectively targeted by our approach.

H23 cells, previously classified as Sotorasib-resistant[52,58], harbor deleterious mutations in tumor suppressor genes such as *STK11* (p.W332Ter), *SMARCA4* (p.K1566_E1567delinsNTer), and *KEAP1* (p.Q193H) (Supplementary Fig. 9B). These co-mutations are associated with poor clinical outcomes in patients treated with KRAS[G12C] inhibitors[35,59]. Genomic characterization of the resistant H23 cell line (H23-R) revealed copy number variations (CNVs), including gains in *MECOM, BCL6,* and *TP63* and losses in *DICER1, XRCC3,* and *AKT1,* but no single-nucleotide variants (SNVs) (Supplementary Fig. 9A–B). These findings further support HiFiCas9-mediated KRAS[G12C] elimination as a promising therapeutic alternative, particularly in cases where Sotorasib resistance is associated with *KEAP1* and other co-mutations.

Conversely, our approach did not show therapeutic efficacy in the Sotorasib-resistant H358-R cells. Genomic characterization of H358-R1 cells identified a frameshift mutation in *CREBBP (NM_004380.3, c.5181_5182delCCinsT)*, CNV gains in *SMARCB1, FGFR4, FLT4, PDGFRB,* and *MAPK1,* and losses in *KEAP1* and *SMARCA4* (Supplementary Fig. 9C–D). These alterations likely contribute to resistance through mechanisms such as RTK activation, MAPK/PI3K pathway reactivation, *KEAP1*-mediated stress response inhibition[60], chromatin accessibility alterations due to *SMARCA4* loss[60], and impaired KRAS acetylation. Additionally, unlike the parental H358 cells, H358-R2 cells lacked the previously observed *KRAS* amplification (8x), which may explain the absence of therapeutic benefit from our gene-editing approach.

Importantly, our data showed no evidence of adaptation or resistance following consecutive treatments of HiFiCas9 RNPs, with no positive selection of indels preserving oncogenic *KRAS* activity. This suggests that HiFiCas9-mediated KRAS[mut] targeting could circumvent the resistance mechanisms observed in Sotorasib-resistant tumors.

Our findings highlight HiFiCas9-mediated KRAS[G12C] targeting as a durable and comprehensive therapeutic strategy, particularly in cases where current inhibitors fail due to compensatory signaling or resistance.

Another important aspect of this therapy is the analysis of the immune components in an in vivo model. NSCLC patients with *KRAS* mutations frequently exhibit high PD-L1 expression and benefit from anti-PD1 therapies[61,62]. Oncogenic KRAS promotes immune evasion by stabilizing PD-L1 mRNA post-transcriptionally, enabling tumors to escape immune surveillance[63]. Notably, *KRAS*-deficient cells fail to evade the host immune system in syngeneic wild-type mice, triggering a strong antitumor response[64]. Furthermore, KRAS[G12C] inhibitors alone do not fully prevent KRAS-driven immune suppression, as Sotorasib has been shown to synergize with anti-PD-1 immunotherapy[28]. These findings underscore the need for future research in immune-competent models to determine whether sustained mutant *KRAS* expression is a key driver of immune evasion, even in the presence of KRAS[mut] inhibitors.

Given that immunotherapy is a first-line treatment for NSCLC and *KRAS* ablation elicits an antitumor immune response[64], our approach may not only enhance immunotherapy efficacy but also serve as a powerful tool for investigating *KRAS*-driven immunosuppression in NSCLC. However, the clinical translation of this approach hinges on overcoming key limitations, particularly in optimizing in vivo CRISPR-Cas9 delivery to maximize its therapeutic potential.

Despite these challenges, CRISPR-Cas technologies are making strides in clinical oncology. For instance, the NCT02793856 study, which evaluates CRISPR-engineered PD-1 knockout T cells in metastatic NSCLC, highlights the growing potential of gene-editing therapies in lung cancer. With continued advancements in delivery methods, HiFiCas9-mediated KRAS[G12C] targeting could represent a transformative approach to treating *KRAS*-mutant NSCLC.

Our results demonstrate that the HiFiCas9-mediated *KRAS* targeting system can selectively eliminate tumor cells by targeting KRAS[G12C] and KRAS[G12D] driver mutations, highlighting its therapeutic potential in preclinical models. Notably, in PDXO models, HiFiCas9 system exhibited superior inhibition of KRAS-driven oncogenesis compared to Sotorasib. In vivo studies conducted in CDX and PDX models further underscore its efficacy, suggesting that enhanced delivery strategies could maximize its therapeutic impact in future applications. Overall, the HiFiCas9 system presents a valuable tool for investigating KRAS-driven immune suppression, with critical implications for adapting this targeted therapy as a first-line treatment for NSCLC patients.

## Methods

### Ethics declarations

All experimental procedures described in this study were conducted in accordance with relevant institutional and international ethical guidelines and regulations.

For the CDX experiments, protocols were approved by the Animal Experimentation Ethics Committee (CEEA) of the University of Granada and conducted in accordance with the guidelines of the University's Bioethics Committee and the *Guide for the Care and Use of Laboratory Animals*. All procedures were performed under isoflurane inhalation anesthesia, and every effort was made to minimize suffering. Mice were kept in cages under a 12-h light/dark cycle, 20-24 °C, 45-65% relative humidity with food and water available ad libitum. No more than five mice were housed in each cage.

The CrowBio PDX experiments were performed under the Institutional Animal Care and Use Committee (IACUC)-approved protocol CBSD-ACUP-001. All procedures complied with the U.S. Department of Agriculture's Animal Welfare Act (9 CFR Parts 1, 2, and 3), the Association for Assessment and Accreditation of Laboratory Animal Care (AAALAC) standards, and Crown Bioscience San Diego's internal Standard Operating Procedures.

For the PDXO models, PDXs were established at the Seville Institute of Biomedicine (IBIS) and Spanish National Cancer Research Center (CNIO) using lung cancer samples obtained from patients at Virgen del Rocío Hospital (Seville) and 12 de Octubre Hospital (Madrid). The research project was approved by the hospitals' ethics committees (Approval ID:2012PI/241 (Seville) and CEIm 20/090 (Madrid)). All patients provided written informed consent in accordance with the protocols approved by the respective local ethics committee. Animal studies were conducted in compliance with established animal care guidelines and were approved by the Consejería de Agricultura of the Junta de Andalucía (Approval ref: SSA/SI/MD/pdm) and by the Animal Protection Office of the Comunidad Autónoma de Madrid (Approval ID: PROEX 084/15, PROEX 313/19, PROEX 297.5/22).

Selection of human PDXs, PDXOs, or derived cell lines for this study was based on *KRAS* mutational status and not on the sex/gender or race/ethnicity of the patient of origin, as there is no evidence of sex-related differences in the molecular mechanisms driven by oncogenic *KRAS*.

### Cell lines

Eight NSCLC cell lines (NCI-H1299, #CRL-5803; NCI-H358, #CRL-5807; NCI-H23, #CRL-5800; NCI-H1792, #CRL-5895; NCI-H2122, #CRL-5985; NCI-A427, #HTB-53; NCI-H838, #CRL-5844; and SKLU-1, #HTB-57) were purchased from the American Type Culture Collection (ATCC). All cell lines were tested for mycoplasma contamination prior to their use. Cells were cultured in RPMI 1640 medium (Biowest, #L0501), supplemented with 10% FBS, 1% penicillin–streptomycin, and 1% L-Glutamine.

RAS-less MEFs were first developed by Mattias Drosten, Mariano Barbacid and colleagues[11]. The RAS initiative team at the Frederik National Laboratory for Cancer Research expanded their work afterwards, and they generated a panel of isogenic MEF cell lines carrying different variants of the human *KRAS*. RAS initiative team kindly donated us the *KRAS^WT* (#RPZ25854), *KRAS^G12C* (#RPZ26186), and *KRAS^G12D* (#RPZ26198) MEF variants (https://www.cancer.gov/research/key-initiatives/ras/ras-central/blog/2017/rasless-mefs-drug-screens). These cells were cultured in DMEM High Glucose medium (Thermo-Fisher, #11995040) + 10% FBS.

## Sotorasib-resistant cell lines
Sotorasib-resistant cell lines were developed by Eloisa Jantus-Lewintre's group (H358-R1 and resistant H23 cell line) and Irene Ferrer's group (H358-R2). Resistance to sotorasib was induced through incremental exposure, with stepwise dose escalation up to 5 μM (H358-R1), 2 μM (H358-R2) or 10 μM (H23).

## 3D spheroid cultures
To culture lung cancer cells as 3D spheroids, we used ultra-low attachment plates (Corning Costar®, #3474). Cells were mixed with 50 μL Matrigel (Corning, #354248) in RPMI 1640 growth medium in a 1:1 proportion to establish the scaffold system for the spheroid suspensions. Then, 100 μL of RPMI medium was added to each well to reach a final volume of 200 μL.

## Cell-derived xenografts
Cell-derived xenograft mouse models of human lung cancer tumors were carried out by implanting GFP/HiFiCas9-AdV-infected H358 or A427 cells (1×10^6 cells in 150 μL of 1:1 Matrigel:RPMI) through subcutaneous injection under rear flanks of 6-to-8-week-old male and female NOD Scid Gamma (NSG) mice. These were obtained from Charles River Laboratories (Barcelona, Spain), originally sourced from The Jackson Laboratory (Bar Harbor, ME, USA). The animals underwent a 14-day quarantine period and were then transferred to the animal housing area. Tumor growth inhibition (TGI) was evaluated twice a week by measuring tumor sizes and volumes using the formula: (1) $V = \frac{1}{2}ab^2$, where $a$ is the tumor's longer axis, and $b$ is the shorter axis. The study was terminated on day 60 or when the combined tumor volume (left + right) reached 3,000 mm³.

## Next-generation sequencing and mutation analysis
**Library construction, quality control and sequencing.** Genomic DNA was extracted using the QuickExtract™ DNA Extraction Solution (Lucigen, #QE09050). Sequencing libraries were generated, and sample-specific indexes were incorporated. Targeted regions were amplified using specific primers (*KRAS*_FW: 5′ TGTATCAAAGAATG GTCCTGCAC 3′; *KRAS*_RV: GATACACGTCTGCAGTCAACT), including barcodes. PCR amplification was conducted with Phusion High-Fidelity DNA Polymerase (ThermoFisher Scientific, #F530S) using the following protocol: 5 min at 95 °C; 35 cycles of (30 s at 95 °C, 30 s at 60 °C, 30 s at 72 °C); 5 min at 72 °C. PCR products with proper size were selected by 2% agarose gel electrophoresis. Equal amounts of PCR products from each sample were pooled, end-repaired, A-tailed, and ligated with Illumina adapters. Sequencing was performed on a paired-end Illumina platform, generating 250 bp paired-end raw reads.

Library quality and quantity were assessed using qPCR. Quantified libraries were pooled and sequenced on the Illumina NovaSeq6000 platform, aiming for a minimum of 30,000 paired-end reads per sample.

**Bioinformatics Analysis Pipeline.** Paired-end reads were demultiplexed by sample barcodes, followed by trimming of barcode and primer sequences. Overlapping paired-end reads were merged using FLASH[65] and further processed for quality filtering with fastp software[66].

Quantification of *KRAS* editing from preprocessed FASTQ files was performed with CRISPResso2[67] in single-end mode with default amplicon end trimming disabled. Allele edition frequencies were recalculated in R version 4.0.2, including only those edits supported by at least 10 reads. Reads tagged as "ambiguous" by CRISPResso2 were classified as edited mutant alleles, as the loss of codon 12 is primarily attributable to editing events specific to mutant alleles, with negligible occurrences in *KRAS* WT cell lines.

## Ribonucleoprotein transfection
CRISPR RNA (crRNA) for *KRAS^G12C* or *KRAS^G12D*, trans-activating crRNA (tracrRNA) and a High-Fidelity version of Cas9 (HiFiCas9) were purchased from Integrated DNA Technologies (IDT), and then properly conjugated to form ribonucleoproteins (RNPs) according to the manufacturer's instructions. HiFiCas9 possesses a single point mutation (p.R691A) that retains high on-target activity of the Cas9 while having reduced off-target activity[19].

## Adenovirus transduction
Adenoviral vectors (AdV) were designed to contain GFP-sgRNAs-*KRAS^G12C*-*KRAS^G12D* HiFiCas9 (AdV-HiFiCas9) or GFP as a control (AdV-Control) (Fig. S5A) and were acquired from VectorBuilder. Cells (300,000 per well) were infected using a multiplicity of infection (MOI) of 1,000 viral particles/cell. Cell suspension was incubated with either AdV-Control or AdV-HiFiCas9 in a 1.5 mL tube for 2 h and then seeded in a 6-well plate.

## Cell viability assay
For comparison of anti-growth activity after gene editing, CellTiter-Glo luminescent-based assay was used (Promega, #G7571). Cells were lipofected with RNPs or infected with AdVs and seeded (3,000 per well) in standard tissue 96-well culture plates (Corning Costar ®, #3903) or ultra-low attachment surface 96-well format plates (Corning Costar ®, #3474), for 2D and 3D culture, respectively. Cell viability was monitored 7 (2D culture) or 12 days (3D culture) later. According to the manufacturer's instructions, 100 μL of CellTiter-Glo reagent was added, vigorously mixed for 5 min, covered, and placed on a plate shaker for 25 min to ensure complete cell lysis prior to assessment of luminescent signal. Luminescence was then measured at 525 nm.

## Protein analysis
Total protein was extracted from the cells using RIPA lysis buffer containing phosphatase and protease inhibitor cocktails (Thermo-Fisher Scientific, #A32955), and then quantified by a Bradford assay. Proteins were then separated by molecular weight and detected by Western blot using anti-KRAS (Santa Cruz, #SC-30), anti-KRAS-G12D (Cell Signaling, #14429S), anti-phERK (Cell Signaling, #4370 T), anti-phAKT (Cell Signaling, #4060 T), anti-phP70S6 (Santa Cruz, #SC-8416), anti-β-actin (Merck, #A5441), and anti-HSP90 (Cell Signaling, #4877) antibodies.

## Immunohistochemistry
Once tumors were fixed in 4% formaldehyde, they were paraffin-embedded and 3-μm-thick sections were obtained. Ki67 (SP6 Rabbit Recombinant Monoclonal (ab21700), Abcam) and Cleaved-Caspase 3 (Rabbit Polyclonal (ab52294), Abcam) immunostainings were performed using standard procedures by AtrysHealth SA (Barcelona, Spain). Representative images were taken using Olympus BX43 microscope (Olympus Life Science, MA, USA). Quantification analyses of stains were carried out with ImageJ Software. Statistical analysis was conducted through Student's t-test with GraphPad 9.

## Patient-derived xenografts

Patient-derived xenografts (PDX) experiments were performed by Crown Bioscience San Diego, California, USA. Selected models were revived as follows; (i) frozen tumor chunks were thawed in a 37 °C water bath, (ii) five female NSG mice were inoculated with tumor chunks on both front flanks, (iii) once the tumor volume reached 700 to 1000 mm³, warm tumors were harvested and processed for tumor chunk inoculation. After harvesting the warm tumors, the tumor was placed in a petri dish over ice in a biosafety cabinet and washed with sterile cold PBS. Connective tissue, fat, excess skin, and necrotic tissue were removed. Using a sterile scalpel, the tumors were sliced into 2-3 mm squared tumor fragments. The tumor fragments were transferred to a petri dish over ice and washed in cold PBS. A sterile trocar was loaded with a 2×2 mm tumor fragment using forceps. The mice were anesthetized using isoflurane for tumor implantation (LU5245 ($KRAS^{G12C}$) and LU5162 ($KRAS^{G12D}$)). When the average tumor volume reached 150-200 mm³ (if tumor volume from one flank reached 150-200 mm³) mice were randomly assigned to the respective treatment groups (AdV-GFP (control) or AdV-HiFiCas9 (treatment)) and dosed within 24 h of randomization. Animals were treated with $10^{11}$ virus particles (corresponding to 50 µl from the AdV vials) administered in 3 intratumoral injections every 2-3 days. The study was terminated on day 28 or when the combined tumor volume (left + right) reached 3,000 mm³. Tumor volume was calculated using the following formula:

$$V = \frac{Longest\ diameter * Shortest\ diameter^2}{2} \quad (1)$$

Tumors and body weights were measured 2 times per week for the duration of the study. The analysis of the differential growth of PDX tumors over time was largely based on the methodology described by Oberg et al[68]. Tumor growth was modeled using the following mixed effects model:

$$\log_2(V) = \beta_0 + \beta_1 trx + \beta_2 t_{mid} + \beta_3 t_{mid} trx + \varepsilon_{tumor} + \varepsilon_i \quad (2)$$

Where $V$ was the tumor volume (in mm²), $trx$ codified the treatment (control = 0, Cas9 = 1), $t_{mid}$ was the time (in days) centered around the midpoint of the experiment (day 13), and $\varepsilon_{tumor}$ was a random intercept term that depended on each tumor. We checked that the starting tumor sizes were independent from each other and that treatment-independent differences in the tumor growth rate were negligible. Furthermore, given the limited number of data points, we corroborated that a straight model with no time-dependent variance-covariance structure was most appropriate. We tested the goodness of fit of various alternative models based on the Akaike information criterion, the Bayesian information criterion, residual plots, and plotting the models vs. the actual data. The differences in tumor growth rate between treatments were tested based on whether the $\beta_3$ coefficients were significantly different from 0 under a $t$ distribution. For all analyses, R version 4.0.2 was used, and the nlme package (version 3.1-152) was used for constructing the models.

## PDX-derived organoids (PDXOs)

PDX tumors (from PDX collection of Irene Ferrer and Luis Paz-Ares´ groups), derived from patient biopsies and grown in immunocompromised mice, were enzymatically digested with collagenase (1.2 mg/ml, #C9891, Sigma), DNase (10 mg/ml, #D5025, Sigma), dispase (0.125 mg/ml, #17105041, Gibco) in Basic medium (Advanced DMEM/F12 (#12634010, Gibco), 1x HEPES (#15630106, Gibco), 1x Glutamine (#G8541, Sigma)) at 37 °C in agitation for 1 h. After incubation, the digested tumor pieces were filtered using 70 µm filters, and the disaggregated cells were centrifuged at 200 x g for 5 min. After two washes with Basic medium, live cells were counted. 5×10⁶ cells/well in 2 ml of Basic medium were plated in a 6-well Ultra-Low-Attachment (ULA) plate (#3471, Corning). To infect the cells,

1000 virus particles (VP) of AdV-GFP and AdV-HiFiCas9 per cell and 8 µg/ml of polybrene were added to the well. Cells were centrifuged at 600 x g 30 °C for 1 h. Then, the plate was incubated at 37 °C 5% $CO_2$ incubator for 3-4 h. After incubation, infected cells were centrifuged at 200 x g for 5 min. A triplicate of 5000 cells/well per condition were resuspended in 36 µl/well of Matrigel (phenol red free and growth factor reduced, #CLS356231, Corning) in a 96-well plate. Cells in Matrigel were allowed to solidify at 37 °C, 5% $CO_2$ in incubator for 10 min and 150 µl/well of Complete medium (Basic Medium supplemented with 2% FBS, 3 ng/ml epidermoid growth factor (EGF) (#PMG8041, Life Technologies), 5 mg/ml human insulin (#I3536, Sigma), 1 mg/ml hydrocortisone (#H0888, Sigma), 1x B27 (#17504-044, Life technologies), 10.5 µM (Y-27632) ROCK inhibitor (#SCM075, Sigma).

Viability assay was carried out with GFP (AdV-Control) and Cas9 (AdV-HiFiCas9) conditions, with and without Sotorasib treatment. Sotorasib 1 µM (#HY-114277, MedChemExpress) was added to the treated condition and refreshed at day 4. After 24 h and 7 days of the infection, cell viability was measured using CellTiter-Glo as previously described. Viability at day 7 was normalized to 24 h' time point and represented as ratio to AdV-Control.

## Statistics & reproducibility

Data are presented as mean values ± SD (standard deviation) or ±SEM (standard error of the mean), calculated using GraphPad Prism v9.0 and R version 4.0.2. P value < 0.05 was considered the threshold for statistical significance. P values are provided within each figure legend, together with the statistical test performed for each experiment: two-tailed one-sample, unpaired two-tailed Student's t-test or two-way ANOVA followed by post hoc test for multiple comparisons calculated in GraphPad Prism v9.0. $N$ values are also indicated within figure legends and refer to biological replicates. Derived statistics correspond to analysis of averaged values across biological replicates, and not pooled technically and biological replicates.

## Reporting summary

Further information on research design is available in the Nature Portfolio Reporting Summary linked to this article.

## Data availability

The data generated *in* this study are provided *in* the Supplementary Information/Source Data file. Raw targeted-sequencing data generated in this study have been deposited into the SRA database under accession number PRJNA965929 (Characterization of indels in Cas9-induced KRAS edition in NSCLC cells) without access restrictions. Source data are provided with this paper.

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

## Acknowledgements

P.P.M.'s laboratory is supported by the grants PID2021-126111OB-I00 and PID2024-159252OB-I00, funded by the MCIN/AEI/10.13039/501100011033 and by ERDF ("A way to make Europe"), the Spanish Association Against Cancer (LAB-AECC-2018), the Junta de Andalucía (PI-0203-2022 and PI-0228-2024), and the University of Granada (B-CTS-480-UGR20, C-EXP-051-UGR23, C-CTS-149-UGR23). J.S.-H. is supported by a grant from the Scientific Foundation of the Spanish Association Against Cancer in Granada (#PRDGR21428SANJ). J.C.A.-P. was supported by an MSCA-IF-EF-RI 837897 fellowship (Horizon 2020 Framework Programme). A.M.A. was supported by an FPU17/01258 fellowship. A.A. was supported by an FPU17/00067 fellowship (Spanish Ministry of Science, Innovation, and Universities) and by the Programa de Contratos Puente (Plan Propio 2022) of the University of Granada. M.S.B.-C. was supported by an FPU19/00576 fellowship (Spanish Ministry of Science, Innovation, and Universities). E.J.-L.'s laboratory is funded by the Spanish Association Against Cancer (PROYE18012ROSE), the Instituto de Salud Carlos III (PI22/01221), and CIBERONC (CB16/12/00350). We would like to thank Ms. Rosario Martín from Écija for her donation to promote research into lung cancer. The authors acknowledge the support of the ROLUCAN Association (Rota Lucha Contra el Cancer) for facilitating this research, and the Ph.D program of Biochemistry and Molecular Biology of the University of Granada.

## Author contributions

P.P.M. conceived the study and allocated the funding for the project; P.P.M. and J.C.A-P coordinated the scientific team and supervised the project, J.S-H., J.C.A-P, and A.M.A. generated most of the experimental data; M.S.B-C and A.A. performed statistical data analyses; I.H-N., I.F. and L.P-A. PDXOs generated resistant cell lines and performed PDXO experiments; S.C-F, and E.J-L provided resistant cell lines; All authors participated in writing and approving the final version of the manuscript.

## Competing interests

The authors declare no competing interests.

## Additional information

[1]Gene Expression Regulation and Cancer Group (CTS-993). GENYO. Centre for Genomics and Oncological Research: Pfizer-University of Granada-Andalusian Regional Government, Granada, Spain. [2]Department of Biochemistry and Molecular Biology I, Faculty of Sciences, University of Granada, Granada, Spain. [3]Instituto de Investigación Biosanitaria ibs, Granada, Spain. [4]Targeted Therapies for Precision Oncology, Instituto de Investigación Hospital 12 de Octubre (i+12), CIBERONC, Madrid, Spain. [5]H12O-CNIO Lung Cancer Clinical Research Unit, Instituto de Investigación Hospital 12 de Octubre (i+12) & Centro Nacional

de Investigaciones Oncológicas (CNIO), CIBERONC, UCM, Madrid, Spain. [6]Department of Biochemistry and Molecular Biology III, Faculty of Medicine, University of Granada, Granada, Spain. [7]Department of Pathology, Universitat de Valencia, Valencia, Spain. [8]Molecular Oncology Lab, Fundación para la Investigación del Hospital General Universitario de Valencia; CIBERONC, Valencia, Spain. [9]Department of Biotechnology, Joint Unit UPV-CIPF Nanomedicine and Disease Mechanisms. Universitat Politècnica de València and Centro de Investigación Príncipe Felipe, Valencia, Spain. [10]These authors contributed equally: Juan Carlos Álvarez-Pérez, Juan Sanjuán-Hidalgo, Alberto M. Arenas. ✉e-mail: pedromedina@ugr.es

