## [Peer Review File · Nature Communications]

High-fidelity Cas9-mediated targeting of KRAS driver mutations restrains lung cancer in preclinical models

Corresponding Author: Professor Pedro P Medina

Version 0:

Reviewer comments:

Reviewer #1

(Remarks to the Author)

Comments to author:

KRAS is a well-known oncogene, and its missense mutations on the 12th amino acid are the major drivers for tumorigenesis and tumor progression. Moreover, CRISPR-Cas9-mediated genome editing has showed clinical potential to suppress KRAS-mutation-driven cell proliferations. In this manuscript, Álvarez-Pérez et al., aimed to use the high-fidelity variants of Cas9 (HiFi-Cas9) to disrupt the G12C and G12D mutants, but not the WT KRAS. With T7EI, ICE, target sequencing, and bioinformatic algorithm, the authors claimed that the well-designed sgRNA combined with HiFi-Cas9 only targeted mutants rather than WT KRAS. Consistently, this strategy only suppressed the growth of cancer cells bearing target mutations in in vitro culture and in xenograft models.

Overall, the topic of this study is interesting, but the conclusions should be strengthened by applying others methods and essential controls.

Major:

1. The author applied HiFi-Cas9-mediated gene editing in different cell lines and used T7EI assay, ICE, and target sequencing to evaluate the editing efficiency. Therefore, the author must show the transfection efficiency and the protein level of HiFi-Cas9 in different cell lines. Moreover, T7EI assay is not sensitive and cannot detect some kinds of small indels. Additionally, targeting sequencing cannot distinguish biological and PCR duplications. Collectively, other high-throughput sequencing methods should be used to assess the editing efficiency on WT and mutated KRAS.
2. Additionally, the same editing strategy should be applied in human cell lines (<https://www.cancer.gov/research/key-initiatives/ras/outreach/reference-reagents/cell-lines>) with KRAS mutation on one allele, while the rest is wildtype. This system should control the differences of transfection and protein level of Cas9.
3. Regarding the off-target activity, the author only used OFFinder and Off-Spotter algorithms, which show a big gap in comparison to the in vivo methods. Thus, it is essential to use high-throughput sequencing methods to evaluate the safety of the editing strategy. Similarly, if HiFi-Cas9 could distinguish a single substitution in the gRNA sequence, it is possible that HiFi-Cas9 may have different off-target site in different individuals or cells.
4. Lots of studies have reported that viral genome can be integrated into the editing site, thus, it is possible that the integration may promote cell proliferation. Regarding this, the author should discuss this possibility.

Minor:

1. Fig. 1c and 1d need more repeats, and the p-value must be shown.
2. The assembly of figure2 is too arbitrary to be followed. 2B should be put in 2D, as these panels must be cited orderly.
3. Figure 2a, 2b, 2c, 2d, and 2f should be displayed as dot-plot showing the individual data, moreover, the biological replicates must be shown as well.
4. Figure 4B, the statistics analysis and relevant p-values must be shown.
5. Figure 4C, scale bar is essential for each picture and the treatment must be clearly labeled.

Reviewer #2

(Remarks to the Author)

The work presented by Alvarez-Peres, Sanjuan-Hidalgo, Arenas et al, revealed an efficient strategy to genetically delete the

oncogenic forms of KRASG12C and KRASG12D. The work has interest and is timely appropriated on regard of the low efficacy of KRASG12C targeted therapy on lung adenocarcinoma patient overall survival.

The CRISPR approach is very elegant, affecting only the KRASG12C KRASG12D oncogene sequences while showing no effect on KRAS WT and this is highly appreciated. Even more, the effect on cell line xenografts (Figure 3) is appealing. On the contrary, the effects on PDX (Figure 4), is at best modest. The finding needs to be confirmed in several others relevant preclinical models, at least *in vitro*, since the delivery of CRISPR *in vivo* so far is complicated. Hence, a way to bypass this problem it could be for instance the use of patient derived organoids.

Even more, sotorasib and/or adagrasib treatments have in general a good first effect, but relapse appears very fast. Hence it will be of interest, again on preclinical models, the combination of both treatments comparing with only the current KRASG12C targeted therapy. Particularly in overall survival of mice where an increased should be expected (in xenografts as in Figure 3 where the adenoviral delivery seems to work better than on PDX). The reason for that is that the mechanisms of resistance to this targeted therapy are associated in general to either further mutations in oncogenic KRASG12C that hamper the interaction of the inhibitor with the GDP binding site on KRAS, or KRAS oncogene amplification, and in both cases, deletion should hamper the appearance of resistance.

Finally, it would be also important to test this approach in already sotorasib/adagrasib resistant KRASG12C cell lines, *i.e.*, in the acquired resistance scenario.

Below I detailed some experiments that I hope it will help to increase the impact of this interesting approach.

- 1) Authors need to demonstrate a real benefit in preclinical models, *i.e.*, at least in 3 different patient-derived organoids. In all these models, authors should treat with KRASG12C targeted therapy or the CRISPR approach alone (monotherapy) but also in combination to validate the clinical impact of this new method.
- 2) I truly believe that is worthy to find another PDX that could have a real benefit *in vivo*, but if results on the point 1) is good it could be enough.
- 3) The characterization of the treated xenografts, either H358 and A427 or the PDX is very low. A full characterization of at least the MAPK pathway should be provided, either by IHC or WB as well as measuring Ki67 and caspase 3 levels on IHC.
- 4) Why the authors did not treat with the KRASG12C targeted therapy *in vivo*? It should be done at least in one of the PDX models, better in another one where the authors might find a better effect than the 2 already shown (please see point 2). In this experiment as proposed in point 1, a comparison with the current KRASG12C targeted therapy should be performed. *i.e.*, perform 3 treatments: KRASG12C targeted therapy or the CRISPR method alone (monotherapy) and also in combination to validate the clinical impact of this new method. As mentioned above, this would increase the quality of the work, but, if the point 1 showed very clear results in the 3 patient-derived organoid models, it could be enough.
- 5) The point of lack of synergy between the CRISPR approach and sotorasib on Figure 3 is well taken. On the contrary, and since the biggest problem of KRASG12C targeted therapy is the development of resistance, authors should test if this approach avoid or at least delay the appearance of resistance against this treatment. Ideally this should be demonstrated in at least 2 cell lines, one KRASG12C and another KRASG12D. These experiments can be performed *in vitro*.
- 6) Authors should enter in contact with labs with already available resistant sotorasib/adagrasib KRASG12C cell lines (or develop them in house), and test for a therapeutic benefit upon genetic deletion with their new approach. At least 2 cell lines should be tested. *In vitro* could be enough (2D and 3D). Like this, they could test for the benefit in acquired resistance.

Reviewer #3

(Remarks to the Author)

In this article, Alvarez-Perez, Sanjuan-Hidalgo, Arenas et al. propose a method to knock out KRAS mutants without affecting KRAS WT and with limited OFF-target. The authors used the previously published system HiFi-Cas9 and tested several sgRNA targeting KRASG12C or KRASG12D in several KRAS mutant and WT cell lines. Using the most specific and efficient sgRNAs, the authors show that KRASmut KO affects cell viability and RAS downstream signalling in 2D and 3D cultures and that KRASmut KO has a superior efficacy than KRASG12C inhibitor sotorasib to prevent clonogenicity in 2D and cell viability. The authors then use *in vivo* models to show that KRASmut KO in human cells before transplantation into immunodeficient mice reduced tumour growth. They finally use PDX, where they inject adenoviruses containing HiFi-Cas9 vector and sgKRASG12C/D in the tumours of tumour-bearing mice and show a modest decrease in tumour growth in the KRASmut KO PDX.

The introduction provides a clear overview of the study, and the discussion highlights relevant points. However, whilst the idea of silencing KRAS mutants sounds promising for KRAS-driven cancer, the presented results do not convincingly demonstrate its superiority over currently available KRASG12C inhibitors (and soon-to-be-available KRASG12D inhibitors). KRASG12C inhibitors are easily administered, highly specific, and generally well-tolerated by patients (except where the patient would have received IO before). The authors compare the efficacy of their system to sotorasib in only two experiments (Figures 2E and F), where one could assume the difference in cell viability between sotorasib treatment (GFP + 6nM) and KRASmutKO (Cas9 + DMSO) is not significant since not notified on the graph. The work would have been more attractive if KRASmut KO showed greater effects than KRASG12C inhibitors in their *in vivo* experiments, justifying the use of this system over small molecules.

Major comments

- The first paragraph in the results section may be challenging for non-experts in CRISPR technology. Could it be revised to improve clarity?
- Figure 1 legend is unclear and lacks information, such as the signification of the coloured frames on the sequences.
- Figure 1: The authors presented the edition efficiency for KRASG12C P1-sgRNA in D using H358 cells whilst showing

H23 cell gels. Showing efficiencies and gels for all used cell lines would be more coherent. Moreover, the efficiencies calculated from the gels shown in C are very low, suggesting differences in editing efficiency across cell lines which the authors need to discuss. It is also unclear why the authors chose the formula they used instead of $100 \times (F1 + F2) / (F1 + F2 + FL)$.

- Figure 1D: Should the experiment be repeated, and the mean efficiency presented with error bars?
- Figures 2A, C and D: Although significant, the effect size on cell viability is small for most cell lines. It could be due to a lower editing efficiency in these cell lines or the fact that not all KRAS mutant cell lines are addicted to RAS signalling (Sing et al., 2009). Also, in my opinion, cell viability should be presented on a linear scale.
- Figure 2F: Is KRASmut KO more efficient in decreasing cell viability than adagrasib? It is a crucial point, in my opinion. The authors should prefer an ANOVA over a t-test in this experiment and each time when comparing multiple conditions.
- Figure 2: the legend should mention the number of independent replicates, whether the mean or median is presented and how were calculated the error bars—same comments for all legends. The timing of treatment should also be mentioned in the legend, and whether the drug was added again in the long experiments.
- Figure 4B: The legend should specify how were the error bars calculated and what statistic test was performed. The tumour volume should be presented in tumour volume (mm^3) instead of normalised to day1 volume to show the variability at the beginning of the experiment. The individual tumour volume over time should be presented in supplementary (one graph per condition).
- Figure 4C: Since the total number of tumours is not too large, all photos should be presented in the figures if available.

Minor comments

- Figure 1C could be separated into more subpanels; one panel per gel would make the text easier to follow.
- In lines 276-277, the viability assays are mentioned in %, but the corresponding figure (Figure 2A) is presented in ratio. The figure should be changed to match the results in the text.

Version 1:

Reviewer comments:

Reviewer #1

(Remarks to the Author)

I appreciate the authors' efforts in conducting extensive experiments and providing relevant analyses, which have addressed my initial concerns.

Reviewer #2

(Remarks to the Author)

Authors made a great effort to answer to all my concerns. I find that the study largely increased its impact as well as its clinical relevance.

Answers to Reviewers' comments:

We sincerely thank the Reviewers for their valuable comments and constructive critiques. We have addressed all concerns and incorporated new data and figures where necessary. Reviewer comments are shown in blue, and our responses are presented in black. In the revised manuscript, all updated text is highlighted in red.

Reviewer #1 (R#1)

KRAS is a well-known oncogene, and its missense mutations on the 12th amino acid are the major drivers for tumorigenesis and tumor progression. Moreover, CRISPR-Cas9-mediated genome editing has showed clinical potential to suppress KRAS-mutation-driven cell proliferations. In this manuscript, Álvarez-Pérez et al., aimed to use the high-fidelity variants of Cas9 (HiFi-Cas9) to disrupt the G12C and G12D mutants, but not the WT KRAS. With T7E1, ICE, target sequencing, and bioinformatic algorithm, the authors claimed that the well-designed sgRNA combined with HiFi-Cas9 only targeted mutants rather than WT KRAS. Consistently, this strategy only suppressed the growth of cancer cells bearing target mutations in *in vitro* culture and in xenograft models.

Overall, the topic of this study is interesting, but the conclusions should be strengthened by applying other methods and essential controls.

We thank the R#1 for the insightful comments that helped us to improve our manuscript. Next, we answer them point by point.

Major points:

R#1.1. The author applied HiFi-Cas9-mediated gene editing in different cell lines and used T7E1 assay, ICE, and target sequencing to evaluate the editing efficiency. Therefore, the author must show the transfection efficiency and the protein level of HiFi-Cas9 in different cell lines. Moreover, T7E1 assay is not sensitive and cannot detect some kinds of small indels. Additionally, targeting sequencing cannot distinguish biological and PCR duplications. Collectively, other high-throughput sequencing methods should be used to assess the editing efficiency on WT and mutated KRAS.

Following R#1 suggestion, we have added data to support our system's transfection and editing efficiency. Our HiFiCas9 system was delivered *in vitro* by transfecting a quantified amount of ribonucleoprotein particles (RNPs). This contrasts with vector-based methods where there is no sustained Cas9 protein production. Consequently, the protein is delivered transiently, rapidly diminishes, and becomes undetectable within 3 to 4 hours because there is no continuous production mechanism^{1,2}. For this reason, the method of measuring the HiFiCas9 protein level must differ from vector-based methods. Instead of using western-blot to quantify the HiFiCas9 levels, we have quantified the uptake of the KRAS^{G12C} sgRNA-containing ribonucleoprotein particles (RNPs) using a tracrRNA labeled with the fluorescent compound ATTO™ 550. As displayed in the Supplementary file 1A (attached below), the RNP transfection efficiency was high (>90%) and comparable between KRAS^{WT} (H838) and KRAS^{G12C} cells (H23).

Supplementary file 1. (A) Transfection efficiency of the different KRAS^{G12C}-mutant-specific RNPs. Flow cytometry data from the transfections involving KRAS^{G12C} sgRNA-containing RNPs, as shown in Figure 1C. The X-axis displays the side scatter values (indicating internal complexity), while the Y-axis shows the ATTO 550 signal (Excitation 553/Emission 575), which is proportional to the internalized RNPs.

Regarding R#1's concerns about assessing editing efficiency, we acknowledge that targeted amplicon + NGS is not suitable for quantifying large deletions or insertions. However, it is the preferred method for accurately measuring editing efficiency. High-throughput sequencing methods are more appropriate for detecting off-target effects, as mentioned by R#1 and discussed in point R#1.3. To better characterize the editing efficiency and specificity of our models, we adjusted sequencing protocols (detailed in the Methods section of the new manuscript version), included biological replicates, and analyzed reads with CRISPResso2³ (<http://crispresso2.pinelloilab.org/submission>) (shown in Figure 1F, previously Figure 1D).

Our targeted NGS analysis yielded several key findings on indel patterns, reinforcing our system's efficacy and specificity: (i) practically all reads from the WT allele remained non-edited with both sgRNAs; KRAS^{G12C} edits were 0.6% in H838 and 0.5% in H358, while KRAS^{G12D} edits were 1% in H838 and 2.1% in A427; (ii) for the mutant allele, KRAS^{G12C}-gRNA edited 65% of reads, and KRAS^{G12D}-gRNA edited 78%; and (iii) indel distribution was highly consistent between replicates and similar in both mutant cell lines (Supplementary File 3).

R#1.2. Additionally, the same editing strategy should be applied in human cell lines (<https://www.cancer.gov/research/key-initiatives/ras/outreach/reference-reagents/cell-lines>) with *KRAS* mutation on one allele, while the rest is wildtype. This system should control the differences of transfection and protein level of Cas9.

We apologize for not making this clear earlier. A427, H23, and H358 are human lung cancer cell lines with heterozygous *KRAS* mutations. Our system was tested in these lines, which include both wildtype and mutant *KRAS* alleles, such as A427 (*KRAS*^{G12D} and *KRAS*^{WT}) and H358 (*KRAS*^{G12C} and *KRAS*^{WT}). The H838 cell line, which has no *KRAS* mutations (*KRAS*^{WT}/*KRAS*^{WT}), was used as a control to test the specificity of our system. The results from the efficiency and specificity analysis of the editing system are detailed in response to R#1.1.

R#1.3. Regarding the off-target activity, the author only used OFFinder and Off-Spotter algorithms, which show a big gap in comparison to the in vivo methods. Thus, it is essential to use high-throughput sequencing methods to evaluate the safety of the editing strategy. Similarly, if HiFi-Cas9 could distinguish a single substitution in the gRNA sequence, it is possible that HiFi-Cas9 may have different off-target site in different individuals or cells.

We appreciate R#1's concerns about potential off-target effects of HiFiCas9. This technical matter has been comprehensively studied using the high-throughput sequencing method GUIDE-Seq (Genome-wide, Unbiased Identification of DSBs Enabled by Sequencing), which allows for the unbiased detection of off-target genome editing events in DNA caused by editing systems⁴. This analysis concluded that HiFiCas9 and regular Cas9 display the same off-target sites and repair outcomes, but HiFiCas9 exhibited a remarkable reduction in off-target editing⁵. Our findings, focusing on *KRAS*^{WT} as the most evident off-target due to its similarity to *KRAS*^{mut} sequences, align perfectly with these results. No other sequences showed similar homology (only a single nucleotide difference). Remarkably, Cas-OFFinder and Off-Spotter algorithms predicted other sequences with more than one nucleotide difference, which we tested by Sanger sequencing. We have updated the Supplementary Figure 4 improving the presentation of this analysis. Non-HiFiCas9 failed to discriminate single-nucleotide differences, resulting in editing of *KRAS*^{WT} present in both alleles of H1299 and in one of the H23 cell lines (Supplementary File 1B). In contrast, HiFiCas9 exhibited superior specificity without any loss of efficacy at *KRAS*^{mut} on-target sites (Figure 1E).

R#1.4. Lots of studies have reported that viral genome can be integrated into the editing site, thus, it is possible that the integration may promote cell proliferation. Regarding this, the author should discuss this possibility.

We believe the R#1 is referring to the observations made with Adeno-Associated Virus (AAV)-mediated delivery. Following transduction, AAV DNA predominantly remains episomal, with some degree of integration into hotspots in mitochondrial DNA and a specific location on chromosome 19 termed AAVS1⁶. In addition, as the R#1 points out, high levels of AAV integration (up to 47%) into Cas9-induced double-strand breaks (DSBs) have been observed in various tissues of treated mice⁷.

However, we used adenoviral vectors (AdVs) instead of AAV. AdVs lack endogenous machinery and do not integrate into the host genome, thus avoiding the risks of off-target effects and insertional mutagenesis associated with other vectors⁸. To the best of our knowledge, no integration at the editing site has been described when AdVs have been used as delivery vectors for CRISPR components. In fact, AdVs are being used in preclinical studies; for example, AdV-mediated delivery of CRISPR/Cas9 has achieved successfully to target *in vivo* gene alpha-1-antitrypsin⁹. If AdVs were to integrate at the *KRAS* site, which is unlikely, it would disrupt *KRAS* mutant expression, leading to toxicity in tumor cells.

R#1 Minor comments:

1. Fig. 1c and 1d need more repeats, and the p-value must be shown.
2. The assembly of figure2 is too arbitrary to be followed. 2B should be put in 2D, as these panels must be cited orderly.
3. Figure 2a, 2b, 2c, 2d, and 2f should be displayed as dot-plot showing the individual data, moreover, the biological replicates must be shown as well.
4. Figure 4B, the statistics analysis and relevant p-values must be shown.
5. Figure 4C, scale bar is essential for each picture and the treatment must be clearly labeled.

We appreciate R#1's valuable suggestions, which have improved our manuscript's clarity. We have implemented accordingly the following modifications:

1. Following the request of R#1, we added more replicates to Figure 1F (previously Figure 1D) and included p-value.

2. We appreciate R#1 suggestion. To address concerns about assembly clarity, we have modified and reorganized the panels in Figure 2 to align with their citation order.

FIGURE 2

Figure 2. Genome editing of $KRAS^{G12C}$ and $KRAS^{G12D}$ impairs viability of tumor cells *in vitro*.

(A) Cell viability of $KRAS^{mut}$ NSCLC cell lines targeted with mutation-specific RNPs, 7 days after transfection (N=3-6).

(B) 2D cell viability assay of NSCLC cell lines 10 days after AdV transduction (N=5-7).

(C) Left: 3D cell viability assay of NSCLC cell lines 10 days after AdV transduction (N=6-7). Right: representative 3D spheroid culture of H358 cells.

(D) Western Blots displaying $KRAS^{G12C}$ -dependent signaling 72h after RNP transfection with densitometry quantification of three independent experiments (Left: H358; Right: A427, N=3).

Statistical analysis: One sample t-test (panels A-D).

(E) Western Blot displaying $KRAS^{G12D}$ -dependent signaling in H358 cells 72h after AdV transduction.

(Representative image of N=3).

Bars represent mean values with standard deviation (SD). Treatment was administered once at time point $t=0$.

3. The graphs in Figure 2 have been redesigned, and additional information about replicates has been included in the figure's legend.

FIGURE 2

Figure 2. Genome editing of $KRAS^{G12C}$ and $KRAS^{G12D}$ impairs viability of tumor cells *in vitro*.

(A) Cell viability of $KRAS^{mut}$ NSCLC cell lines targeted with mutation-specific RNPs, 7 days after transfection (N=3-6).

(B) 2D cell viability assay of NSCLC cell lines 10 days after AdV transduction (N=5-7).

(C) Left: 3D cell viability assay of NSCLC cell lines 10 days after AdV transduction (N=6-7). Right: representative 3D spheroid culture of H358 cells.

(D) Western Blots displaying KRAS-dependent signaling 72h after RNP transfection with densitometry quantification of three independent experiments (Left: H358; Right: A427, N=3).

Statistical analysis: One sample t-test (panels A-D).

(E) Western Blot displaying KRAS-dependent signaling in H358 cells 72h after AdV transduction.

(Representative image of N=3).

Bars represent mean values with standard deviation (SD). Treatment was administered once at time point t=0.

4. The statistical analysis performed for the data presented now in Figure 3D (previously Figure 4B) is now summarized in the "Materials and Methods" section (lines 222 to 237). Additionally, p-values are now explicitly reported in Figure 3D.

FIGURE 3

5. To enhance clarity and compare tumor sizes, we have added scale bars and tumor volumes to Figure 3E (previously Figure 4C).

FIGURE 3

Figure 3. Edition of mutant KRAS induces tumor growth inhibition in PDX and CDXs models.

- (A) Schematic representation of the experimental design with representative *ex vivo* images of tumors harvested 60 days after implantation. 10⁶ pretreated cells were transplanted subcutaneously and tumor growth was monitored for two months. Representative images of N=9.
- (B) Quantification of *ex vivo* tumor volumes of H358 and A427 CDXs (N=9). 63% reduction in tumor volume for H358 (left; p-value: 0.00017) and a ~42% reduction for A427 (right; p-value: 0.015).
- (C) Schematic representation of PDX experimental design.
- (D) Tumor volumes of LU5245 (G12C; left, N=6) and LU5162 (G12D, right, N=4) PDXs. Mean tumor volume (mm³) normalized to day 1, accompanied by standard deviation. Tumor growth was modeled using a linear mixed effects models with random intercepts.
- (E) *Ex vivo* pictures from KRAS^{G12C} tumors extracted 28 days post-treatment. Scale bar=10 mm. Diameters (mm): PDX1: Control=14, HiFi-Cas9=11; PDX2: Control=13.6, HiFi-Cas9=11.8. Volumes (mm³): PDX1: Control=1084, HiFi-Cas9=548; PDX2: Control=948, HiFi-Cas9=658.

Reviewer #2 (R#2)

The work presented by Alvarez-Peres, Sanjuan-Hidalgo, Arenas et al, revealed an efficient strategy to genetically delete the oncogenic forms of *KRAS*^{G12C} and *KRAS*^{G12D}. The work has interest and is timely appropriated on regard of the low efficacy of *KRAS*^{G12C} targeted therapy on lung adenocarcinoma patient overall survival.

The CRISPR approach is very elegant, affecting only the *KRAS*^{G12C} *KRAS*^{G12D} oncogene sequences while showing no effect on *KRAS*^{WT} and this is highly appreciated. Even more, the effect on cell line xenografts (Figure 3) is appealing. On the contrary, the effects on PDX (Figure 4), is at best modest. The finding needs to be confirmed in several others relevant preclinical models, at least in vitro, since the delivery of CRISPR in vivo so far is complicated. Hence, a way to bypass this problem it could be for instance the use of patient derived organoids. Even more, sotorasib and/or adagrasib treatments have in general a good first effect, but relapse appears very fast. Hence it will be of interest, again on preclinical models, the combination of both treatments comparing with only the current *KRAS*^{G12C} targeted therapy. Particularly in overall survival of mice where an increased should be expected (in xenografts as in Figure 3 where the adenoviral delivery seems to work better than on PDX). The reason for that is that the mechanisms of resistance to this targeted therapy are associated in general to either further mutations in oncogenic *KRAS*^{G12C} that hamper the interaction of the inhibitor with the GDP binding site on *KRAS*, or *KRAS* oncogene amplification, and in both cases, deletion should hamper the appearance of resistance. Finally, it would be also important to test this approach in already sotorasib/adagrasib resistant *KRAS*^{G12C} cell lines, i.e., in the acquired resistance scenario.

We sincerely appreciate Reviewer #2's insightful comments on our manuscript, which we believe have significantly enhanced our study. Below, we provide a point-by-point response.

Major points:

R#2.1. Authors need to demonstrate a real benefit in preclinical models, i.e., at least in 3 different patient-derived organoids. In all these models, authors should treat with *KRAS*^{G12C} targeted therapy or the CRISPR approach alone (monotherapy) but also in combination to validate the clinical impact of this new method.

We fully acknowledge and appreciate Reviewer #2's emphasis on clinical impact. We are grateful for the suggestion to utilize organoids as an experimental platform, given the current limitations of in vivo delivery. However, working with organoids has been particularly challenging for our team, as we lacked a collection of *KRAS*-mutant organoids and had to develop specific protocols for delivering our HiFiCas9 system. To overcome this, we established a collaboration with Dr. Paz-Ares' group (CNIO, Madrid), which maintains a valuable collection of PDX-derived organoid (PDXO) models, including some harboring the *KRAS*^{G12C} mutation. We leveraged these models to evaluate our therapeutic strategy.

We selected three Non-Small Cell Lung Cancer (NSCLC) PDXO models with heterozygous *KRAS*^{G12C/WT} mutations: TP181, TP79, and TP60 (Figure 4C). Following Reviewer #2's

suggestion, we compared the effects of genomic elimination of the *KRAS^{G12C}*-allele with Sotorasib treatment. The experimental conditions included AdV-Control, AdV-HiFiCas9, AdV-Control + Sotorasib, and AdV-HiFiCas9 + Sotorasib. Cell viability was assessed on day 1 (t=1) and day 7 (t=7) using the CellTiter-Glo assay, with data normalized to t=1 and AdV-Control. GFP fluorescence, encoded in the plasmid backbone, confirmed successful transduction of the three PDXO models (Figure 4D).

Cell viability analysis revealed a significant ~30% reduction in TP79 and TP60 organoids following AdV-HiFiCas9 treatment compared to AdV-Control organoids (Fig. 4C). Consistent with our cell line data, the combined treatment (AdV-HiFiCas9 + Sotorasib) did not produce an additional reduction in viability beyond Adv-HiFiCas9 alone. Notably, TP60 responded significantly to both Sotorasib and AdV-HiFiCas9, while TP79 responded exclusively to Adv-HiFiCas9, highlighting the added therapeutic advantage of our system over Sotorasib. In contrast, TP181 showed no significant viability changes across treatment condition (Fig. 4C), suggesting that this model may be independent of *KRAS^{G12C}* activity. Visual inspection further confirmed a marked decrease in organoid size in TP79 and TP60 following AdV-HiFiCas9 treatment, while TP181 remained unaffected (Fig. 4D).

These three PDXO models may represent distinct clinical scenarios: (i) *KRAS^{G12C}*-independent tumors (TP181), (ii) tumors with high *KRAS^{G12C}*-GTP activity that are not sensitive to Sotorasib¹⁰ but sensitive to *KRAS^{G12C}* DNA elimination (TP79), and (iii) highly *KRAS^{G12C}*-dependent tumors (TP60). Notably, the latter scenario aligns with findings from the CodeBreakK 100 trial, which reported a 32% response rate to Sotorasib in NSCLC patients¹¹. Overall, our findings underscore the therapeutic potential of the *KRAS^{mut}*-CRISPR/HiFiCas9 system for treating *KRAS*-mutant tumors, particularly those resistant to current *KRAS^{G12C}* inhibitors.

FIGURE 4

Figure 4. Comparison of Sotorasib and HiFi-Cas9 therapy.

(A) Cell viability assay combining HiFiCas9 therapy and Sotorasib in parental H358 cells (N=3). Control/HiFiCas9 infection was performed 24 hours before Sotorasib treatment. *:p<0.05; **:p<0.01; ***:p<0.001. Statistical analysis: two-way ANOVA.

(B) Colony formation assay combining HiFiCas9 therapy and Sotorasib in parental H358 cells.

(C) Cell viability of PDXOs treated with HiFiCas9 therapy and/or Sotorasib.

(D) Representative images of organoid cultures five days post-treatment with adenovirus control or HiFiCas9.

(E) Cell viability of Sotorasib-resistant cells treated with HiFi-Cas9 therapy in 2D cultures (N=3).

(F) Cell viability of Sotorasib-resistant cells treated with HiFi-Cas9 therapy in 3D cultures (N=3).

(G) Representative images of 3D cultures of cells ten days post-treatment with Adenovirus control or HiFiCas9.

R#2.2. I truly believe that is worthy to find another PDX that could have a real benefit in vivo, but if results on the point 1) is good it could be enough.

Despite the well-documented limitations of in vivo delivery of customized adenoviral vectors (AdVs) into solid tumors¹²⁻¹⁴, we observed a significant tumor growth inhibition (TGI) in PDX1 and PDX2 following treatment with our HiFiCas9 system. We anticipate that therapeutic efficacy could be substantially enhanced with an optimized delivery method, as suggested by our CDX models, which were treated ex vivo (Figure 3A and 3B). However, given the lack of alternative strategies for CRISPR tool delivery, we followed Reviewer #2.1's recommendation and focused our experimental efforts on testing the HiFiCas9 system in three distinct *KRAS^{mut}* PDXO models, where we achieved significant results, as detailed in the previous section.

R#2.3. The characterization of the treated xenografts, either H358 and A427 or the PDX is very low. A full characterization of at least the MAPK pathway should be provided, either by IHC or WB as well as measuring Ki67 and caspase 3 levels on IHC.

Following Reviewer #2's suggestion, we performed a more detailed characterization of the treated PDXs by analyzing Ki67 and Cleaved Caspase-3 expression through immunohistochemistry. As shown in Supplementary File 7B, the levels of both proliferation (Ki67) and apoptosis (Cleaved Caspase-3) markers remained unchanged. This is likely due to the elimination of the edited cells, which led to tumor reduction, while the remaining cells at this time point (28 days post-treatment) primarily represent the unedited population.

Next, we characterized the MAPK pathway in the *KRAS*^{G12C/D} CDX models by assessing phosphorylated MEK and ERK levels (Supplementary File 7A). However, we did not observe a clear or consistent pattern, as the variations may be attributed to tumor-specific differences in proliferation.

This supports the rationale that tumor reduction was primarily driven by the elimination of edited cells, while the residual tumor mass (observed at 28 days in PDX models and 60 days in CDX models) consisted largely of non-transduced, unedited cells.

Supplementary file 7. Xenografts characterization.

- (A) *Ex vivo* immunoblots of $KRAS^{G12C/D}$ CDX tumors extracted 60 days post-treatment.
 (B) Immunohistochemical analysis of $KRAS^{G12C}$ (left) and $KRAS^{G12D}$ (right) PDX models.
 Upper: Cleaved Caspase 3.
 Bottom: Ki67.

R#2.4. Why the authors did not treat with the *KRAS*^{G12C} targeted therapy in vivo? It should be done at least in one of the PDX models, better in another one where the authors might find a better effect than the 2 already shown (please see point 2). In this experiment as proposed in point 1, a comparison with the current *KRAS*^{G12C} targeted therapy should be performed. I.e., perform 3 treatments: *KRAS*^{G12C} targeted therapy or the CRISPR method alone (monotherapy) and also in combination to validate the clinical impact of this new method. As mentioned above, this would increase the quality of the work, but, if the point 1 showed very clear results in the 3 patient-derived organoid models, it could be enough.

We believe that directly comparing the efficacy of Sotorasib and our HifiCas9 system in the PDX model is challenging due to differences in delivery efficiency. Among the experimental approaches suggested by Reviewer #2, we opted to evaluate our system in three distinct patient-derived organoid models, where delivery can be more precisely controlled (detailed in point R#2.1).

R#2.5. The point of lack of synergy between the CRISPR approach and sotorasib in Figure 3 is well taken. On the contrary, and since the biggest problem of $KRAS^{G12C}$ targeted therapy is the development of resistance, authors should test if this approach avoids or at least delays the appearance of resistance against this treatment. Ideally this should be demonstrated in at least 2 cell lines, one $KRAS^{G12C}$ and another $KRAS^{G12D}$. These experiments can be performed in vitro.

Thank you, Reviewer #2, for highlighting this key aspect. Our results indicate that Sotorasib does not significantly reduce the viability of cells previously treated with HiFi-Cas9, supporting R#2’s observation of a lack of synergy. We agree that a major challenge in $KRAS^{G12C}$ -targeted therapy is the development of resistance. To address R#2’s insightful question, we designed an experiment to assess potential resistance or adaptation mechanisms arising from three consecutive treatments with our HiFi-Cas9 therapeutic strategy (Supplementary File 8B).

In this approach, we treated H358 and A427 parental cells with HiFi-Cas9 RNPs (Control/scrambled-gRNA and $KRAS^{G12C/D}$ -gRNA) and measured cell viability after seven days. The $KRAS^{G12C/D}$ -gRNA-treated cells were then allowed to recover and proliferate for approximately 30 days before undergoing a second round of RNP transfection and viability assessment. This process was repeated for a third treatment cycle. Across all three consecutive treatments, we consistently observed a loss of viability, indicating that no resistance mechanisms had emerged under this treatment protocol (Supplementary File 8C).

Furthermore, Sanger sequencing of the three post-treatment batches revealed a uniform distribution of $KRAS$ alterations, with no evidence of positive selection for indels indicative of resistance (Supplementary File 8D). We hypothesize that if resistance or adaptation mechanisms to our strategy were to emerge, they would likely involve the positive selection of indels that maintain oncogenic activity while evading recognition and re-editing by $KRAS^{G12C/D}$ -gRNA.

These findings reinforce the notion that the genetic elimination of $KRAS$ mutants is an effective strategy with fewer opportunities for resistance compared to drugs like Sotorasib.

A

B

C

D

R#2.6. Authors should enter in contact with labs with already available resistant sotorasib/adagrasib *KRAS*^{G12C} cell lines (or develop them in house), and test for a therapeutic benefit upon genetic deletion with their new approach. At least 2 cell lines should be tested. In vitro could be enough (2D and 3D). Like this, they could test for the benefit.

Following Reviewer #2's insightful comment, we collaborated with two research groups that had developed Sotorasib-resistant cell lines: Luis Paz-Ares's group at the H120-CNIO Lung Cancer Clinical Research Unit in Madrid and Eloisa Jantus Lewintre's Molecular Oncology group at the Foundation for Research at General University Hospital of Valencia (FIHGU).

Through these collaborations, we obtained two distinct Sotorasib-resistant H358 cell lines generated independently by these two labs (H358-R1 from Valencia and H358-R2 from Madrid) along with a resistant H23 cell line.

We treated these resistant cell lines with our adenoviral-based HiFi-Cas9 targeted therapy in both 2D-adherent and 3D-spheroids cultures. As shown in figure 4E, no statistically significant differences in cell viability were observed in the 2D cultures. However, in the 3D culture, H23-resistant cells exhibited a statistically significant 60% reduction in viability across six independent experiments (N=6, Figure 4F). Notably, H23 cells harbor co-occurring mutations in the *KEAP1* and *SMARCA4* genes, which are associated with poor clinical outcomes in patients treated with Sotorasib^{11,15} (Supplementary File 9B).

These findings related to the Sotorasib-resistant cell lines are further discussed in the revised version of the manuscript (lines 492 to 504).

Additionally, we characterized indel distribution (Supplementary Figure 3B), infection efficiency (Figure 4G), and *KRAS* elimination in the three cell lines (Supplementary Figure 3C).

Reviewer #3 (R#3)

In this article, Alvarez-Perez, Sanjuan-Hidalgo, Arenas et al. propose a method to knock out *KRAS* mutants without affecting *KRAS* WT and with limited OFF-target. The authors used the previously published system HiFi-Cas9 and tested several sgRNA targeting *KRAS*^{G12C} or *KRAS*^{G12D} in several *KRAS* mutant and WT cell lines. Using the most specific and efficient sgRNAs, the authors show that *KRAS*^{mut} KO affects cell viability and RAS downstream signalling in 2D and 3D cultures and that *KRAS*^{mut} KO has a superior efficacy than KRASG12C inhibitor sotorasib to prevent clonogenicity in 2D and cell viability. The authors then use *in vivo* models to show that *KRAS*^{mut} KO in human cells before transplantation into immunodeficient mice reduced tumour growth. They finally use PDX, where they inject adenoviruses containing HiFi-Cas9 vector and sgKRASG12C/D in the tumours of tumour-bearing mice and show a modest decrease in tumour growth in the *KRAS*^{mut} KO PDX.

The introduction provides a clear overview of the study, and the discussion highlights relevant points. However, whilst the idea of silencing *KRAS* mutants sounds promising for *KRAS*-driven cancer, the presented results do not convincingly demonstrate its superiority over currently available *KRAS*^{G12C} inhibitors (and soon-to-be-available *KRAS*^{G12D} inhibitors). *KRAS*^{G12C} inhibitors are easily administered, highly specific, and generally well-tolerated by patients (except where the patient would have received IO before). The authors compare the efficacy of their system to sotorasib in only two experiments (Figures 2E and F), where one could assume the difference in cell viability between sotorasib treatment (GFP + 6nM) and *KRAS*^{mut} KO (Cas9 + DMSO) is not significant since not notified on the graph. The work would have been more attractive if *KRAS*^{mut} KO showed greater effects than *KRAS*^{G12C} inhibitors in their *in vivo* experiments, justifying the use of this system over small molecules.

We thank referee #3 for their insightful general comments. Below, we provide detailed responses to each point.

Major points:

R#3.1. The first paragraph in the results section may be challenging for non-experts in CRISPR technology. Could it be revised to improve clarity?

We appreciate R#3's suggestion for improvement. In response, we have revised the first paragraph to enhance clarity and readability, following their recommendations.

R#3.2. Figure 1 legend is unclear and lacks information, such as the signification of the coloured frames on the sequences.

We thank R#3 for highlighting the missing information. In response to R#3 comment, we have updated the legend of Figure 1 to include additional details, specifically clarifying the significance of the colored frames in Panel C.

Figure 1. Efficiency and specificity of *KRAS*^{mut}-specific sgRNAs.

(A) Representation of *KRAS*^{WT} and *KRAS*^{mut} alleles at genomic DNA level, with the mutated nucleotide highlighted in red.

(B) Graphical representation of the therapeutic approach: *KRAS* oncogene-addicted cells die off when the *KRAS*^{mut}-specific KO is induced. No effect on *KRAS*^{WT}/non-tumor cells.

(C) sgRNA designs utilizing different protospacer adjacent motifs (red frames for PAM1 and green for PAM2) to target *KRAS*^{mut} alleles. Mutant nucleotides are displayed in red, artificially introduced mismatches in blue, and PAM sequences in orange.

(D) T7-endonuclease assay in *KRAS*^{WT} and *KRAS*^{mut} cell lines. Efficiency calculation of *KRAS*^{mut}-specific sgRNAs on digitized agarose gels after T7-endonuclease assay (shown as percentage below the lane with edition) using the formula:
$$\text{Editing Efficiency}(\%) = 100 \times \left(1 - \frac{(F1 + F2)}{(F1 + F2 + FL)} \right)$$
 F1 and F2 refer to the relative pixel density of fragments 1 and 2; FL refers to the full-length undigested amplicon. Gel images are representative of N=3.

(E) T7-endonuclease assay in *KRAS*^{WT} and *KRAS*^{G12C} cell lines using sgRNAs with single mismatches. C-: negative control (100% complementary dsDNA). C+: positive control (heteroduplex with indels).

(F) *KRAS* allele edition frequency by sgRNA. Proportion of unedited and edited targeted high-throughput sequencing reads from either WT, G12C and G12D alleles in heterozygous (H358, A427) and WT homozygous cell lines (H838). T-test, FDR-corrected p values.

R#3.3. Figure 1: The authors presented the edition efficiency for *KRAS*^{G12C} P1-sgRNA in D using H358 cells whilst showing H23 cell gels. Showing efficiencies and gels for all used cell lines would be more coherent. Moreover, the efficiencies calculated from the gels shown in C are very low, suggesting differences in editing efficiency across cell lines which the authors need to discuss. It is also unclear why the authors chose the formula they used instead of $100 \times (F1 + F2) / (F1 + F2 + FL)$.

We appreciate R#3's insightful suggestion. As previous reports indicate that nuclease activity estimates from T7E1 assays often do not accurately reflect editing activity and can significantly differ from NGS results¹⁶, we sought to enhance the reliability of our measurements. To this end, we performed targeted amplicon sequencing with NGS on two additional samples from each cell line, thereby including two more biological replicates in Figure 1F (formerly Figure 1D).

This updated dataset allowed for a more precise assessment of editing efficiency and indel distribution in the H358 (*KRAS*^{G12C}) and A427 (*KRAS*^{G12D}) cell lines, which were central to our *in vitro* and *in vivo* experiments. T7E1 assays tend to underestimate editing efficiency due to the heterozygous nature of these cell lines (*KRAS*^{mut}/*KRAS*^{WT}). In contrast, NGS-based indel analysis differentiates between wild-type and mutant alleles, yielding a more accurate representation of true indel frequency—typically about double the estimate from T7E1 assays.

In T7E1 assays, both the targeted *KRAS*^{G12C/D} and untargeted *KRAS*^{WT} alleles are PCR-amplified and visualized on an agarose gel, causing the densitometry analysis of the unedited band to include the *KRAS*^{WT} allele and underestimate editing efficiency.

Regarding the formula for calculating editing efficiency based on T7E1 assays ($\%NHEJ \text{ events} = 100 \times [1 - (1 - \text{fraction cleaved})^{(1/2)}]$), our literature review suggests this method was historically used for densitometry-based editing efficiency estimates before CRISPR technology¹⁷. To our knowledge, this formula remains widely accepted for CRISPR-based applications^{18,19}. The formula proposed by the reviewer would likely overestimate editing efficiency.

R#3.4. Figure 1D: Should the experiment be repeated, and the mean efficiency presented with error bars?

We agree with Reviewer #3's observation and have addressed it by increasing the number of replicates and including error bars to strengthen the analysis. This additional data enhances confidence in both the editing efficiency and indel distribution. These updates are now reflected in the revised Figure 1.

Figure 1. Efficiency and specificity of *KRAS*^{mut}-specific sgRNAs.

(A) Representation of *KRAS*^{WT} and *KRAS*^{mut} alleles at genomic DNA level, with the mutated nucleotide highlighted in red.

(B) Graphical representation of the therapeutic approach: *KRAS* oncogene-addicted cells die off when the *KRAS*^{mut}-specific KO is induced. No effect on *KRAS*^{WT}/non-tumor cells.

(C) sgRNA designs utilizing different protospacer adjacent motifs (red frames for PAM1 and green for PAM2) to target *KRAS*^{mut} alleles. Mutant nucleotides are displayed in red, artificially introduced mismatches in blue, and PAM sequences in orange.

(D) T7-endonuclease assay in *KRAS*^{WT} and *KRAS*^{mut} cell lines. Efficiency calculation of *KRAS*^{mut}-specific sgRNAs on digitized agarose gels after T7-endonuclease assay (shown as percentage below the lane with edition) using the formula:

$$\text{Editing Efficiency}(\%) = 100 \times \left(1 - \sqrt{\frac{(F1 + F2)}{(F1 + F2 + FL)}}\right)$$

F1 and F2 refer to the relative pixel density of fragments 1 and 2; FL refers to the full-length undigested amplicon. Gel images are representative of N=3.

(E) T7-endonuclease assay in *KRAS*^{WT} and *KRAS*^{G12C} cell lines using sgRNAs with single mismatches. C-: negative control (100% complementary dsDNA). C+: positive control (heteroduplex with indels).

(F) *KRAS* allele edition frequency by sgRNA. Proportion of unedited and edited targeted high-throughput sequencing reads from either WT, G12C and G12D alleles in heterozygous (H358, A427) and WT homozygous cell lines (H838). T-test, FDR-corrected p values.

R#3.5. Figures 2A, C and D: Although significant, the effect size on cell viability is small for most cell lines. It could be due to a lower editing efficiency in these cell lines or the fact that not all KRAS mutant cell lines are addicted to RAS signalling (Sing et al., 2009). Also, in my opinion, cell viability should be presented on a linear scale.

We thank R#3 for raising this point, which we had not specifically discussed. The variability in cell viability cannot be attributed to editing efficiency, as the RNP-editing rates were similar between cell lines (H358 = 65%, A427 = 78%) and did not correlate with the observed viability differences (H358 = 33%, A427 = 64%). As the reviewer noted, it is well-established that KRAS dependency varies widely among $KRAS^{mut}$ cancer cell lines. For instance, H23 has been classified as KRAS-independent²⁰. Recent studies on $KRAS^{G12C}$ inhibitors further highlighted this variability, with viability assays showing that H358 is one of the most sensitive cell lines, while H23 demonstrates marked tolerance^{18,21,22,23}.

Regarding the presentation of cell viability data, following the R#3 reviewer's suggestion, we have modified panel 2A to display viability as a percentage on a linear scale for enhanced clarity.

R#3.6. Figure 2F: Is KRAS^{mut} KO more efficient in decreasing cell viability than adagrasib? It is a crucial point, in my opinion. The authors should prefer an ANOVA over a t-test in this experiment and each time when comparing multiple conditions.

We acknowledge the importance of comparing HiFiCas9-induced KRAS^{G12C} knockout with inhibitor treatment in terms of viability and we apologize for any lack of clarity in our initial communication. It's worth noting that our data show results for Sotorasib rather than Adagrasib. We believe this might have caused some confusion for the reviewer, as both inhibitors share the same mechanism of action.

As suggested, a two-way ANOVA was conducted to assess the effect of Sotorasib and HiFiCas9 treatment on cell viability. The analysis revealed no statistically significant interaction between these variables [$F(2, 12) = 0.55$, $p = 0.5899$]. Individually, Cas9 treatment had a highly significant impact on viability ($F(1, 12) = 69.55$, $p < 0.0001$), accounting for 72.96% of the total variance. Sotorasib, although explaining only 13.3% of the variance, also showed statistical significance ($F(2, 12) = 6.34$, $p = 0.0132$).

A Tukey post-hoc test revealed significant differences between Control/HiFiCas9 treatment across all Sotorasib concentrations. Interestingly, none of the Sotorasib concentrations significantly further reduced the viability of HiFiCas9-treated cells (Figure 4A-B; 6 nM, $p = 0.5062$). Conversely, Sotorasib at any tested concentration did not preclude a significant therapeutic benefit when combined with HiFiCas9 treatment.

To clarify these findings, additional text has been added to lines 438–348 of the manuscript, with an interpretation of the results provided in lines 449–456.

R#3.7. Figure 2: the legend should mention the number of independent replicates, whether the mean or median is presented and how were calculated the error bars—same comments for all legends. The timing of treatment should also be mentioned in the legend, and whether the drug was added again in the long experiments.

As requested by R#3, we have included the exact sample sizes (*n* values) for each relevant figure panel. Additionally, we have updated all figure legends to provide more detailed and relevant information. We sincerely appreciate the reviewer’s helpful feedback.

FIGURE 2

Figure 2. Genomic editing of *KRAS*^{G12C} and *KRAS*^{G12D} impairs viability of tumor cells *in vitro*.

(A) Cell viability of *KRAS*^{mut} NSCLC cell lines targeted with mutation-specific RNPs, 7 days after transfection (N=3-6).

(B) 2D cell viability assay of NSCLC cell lines 10 days after AdV transduction (N=5-7).

(C) Left: 3D cell viability assay of NSCLC cell lines 10 days after AdV transduction (N=6-7). Right: representative 3D spheroid culture of H358 cells.

(D) Western Blots displaying KRAS-dependent signaling 72h after RNP transfection with densitometry quantification of three independent experiments (Left: H358; Right: A427, N=3).

Statistical analysis: One sample t-test (panels A-D).

(E) Western Blot displaying KRAS-dependent signaling in H358 cells 72h after AdV transduction.

(Representative image of N=3).

Bars represent mean values with standard deviation (SD). Treatment was administered once at time point t=0.

R#3.8. Figure 4B: The legend should specify how were the error bars calculated and what statistic test was performed. The tumour volume should be presented in tumour volume (mm^3) instead of normalised to day1 volume to show the variability at the beginning of the experiment. The individual tumour volume over time should be presented in supplementary (one graph per condition).

Following R#3 suggestion, we have enhanced the clarity of Figure 4B. The graph now displays the mean tumor volume (mm^3) normalized to day 1, with error bars representing the standard deviation (SD). This information has been appropriately updated in Figure 3D (previously Figure 4).

Tumor growth over time was analyzed using linear mixed effects models with random intercepts (refer to Methods). Differences in tumor growth rates between treatments were assessed by testing whether the β_3 coefficients significantly differed from 0 under a t distribution. Our analysis revealed that the log-tumor growth rate over time was significantly lower in AdV-HiFiCas9-treated PDXs compared to AdV-Control for both the G12D ($p = 3.6 \cdot 10^{-2}$) and the G12C ($p = 7.9 \cdot 10^{-5}$) mutations.

In response to the reviewer's request, we have added a new supplementary file, Supplementary File 6A, which provides the individual tumor volumes (mm^3) over time without normalization. This file offers a comprehensive view of the variability observed at the beginning of the experiment.

Figure 3. Edition of mutant KRAS induces tumor growth inhibition in PDX and CDXs models.

A
R#3.9. Figure 4C: Since the total number of tumors is not too large, all photos should be presented in the figures if available.

We agree with the reviewer's comment, and to address this, we have appended an additional file—Supplementary File 6 (Panel B)—to the supplementary material, which contains all the photographs of $KRAS^{G12C}$ tumors as reference.

R#3 Minor comments:

1. Figure 1C could be separated into more subpanels; one panel per gel would make the text easier to follow.

As per the suggestion, we have rearranged the panels in Figure 1 to enhance clarity and ease of comprehension.

Figure 1. Efficiency and specificity of *KRAS*^{mut}-specific sgRNAs.

(A) Representation of *KRAS*^{WT} and *KRAS*^{mut} alleles at genomic DNA level, with the mutated nucleotide highlighted in red.

(B) Graphical representation of the therapeutic approach: *KRAS* oncogene-addicted cells die off when the *KRAS*^{mut}-specific KO is induced. No effect on *KRAS*^{WT}/non-tumor cells.

(C) sgRNA designs utilizing different protospacer adjacent motifs (red frames for PAM1 and green for PAM2) to target *KRAS*^{mut} alleles. Mutant nucleotides are displayed in red, artificially introduced mismatches in blue, and PAM sequences in orange.

(D) T7-endonuclease assay in *KRAS*^{WT} and *KRAS*^{mut} cell lines. Efficiency calculation of *KRAS*^{mut}-specific sgRNAs on digitized agarose gels after T7-endonuclease assay (shown as percentage below the lane with edition) using the formula:

$$\text{Editing Efficiency(\%)} = 100 \times \left(1 - \frac{(F1 + F2)}{(F1 + F2 + FL)} \right)$$
 F1 and F2 refer to the relative pixel density of fragments 1 and 2; FL refers to the full-length undigested amplicon. Gel images are representative of N=3.

(E) T7-endonuclease assay in *KRAS*^{WT} and *KRAS*^{G12C} cell lines using sgRNAs with single mismatches. C-: negative control (100% complementary dsDNA). C+: positive control (heteroduplex with indels).

(F) *KRAS* allele edition frequency by sgRNA. Proportion of unedited and edited targeted high-throughput sequencing reads from either WT, G12C and G12D alleles in heterozygous (H358, A427) and WT homozygous cell lines (H838). T-test, FDR-corrected p values.

2. In lines 276-277, the viability assays are mentioned in %, but the corresponding figure (Figure 2A) is presented in ratio. The figure should be changed to match the results in the text.

Thank you for pointing out this inconsistency. As requested, we have added a new Panel A to Figure 2, which now illustrates cell viability as a percentage. Additionally, while recreating the graph, we identified and corrected certain cell viability percentages, which are now detailed in lines 342-345.

References

1. Kim S, Kim D, Cho SW, Kim J, Kim JS. Highly efficient RNA-guided genome editing in human cells via delivery of purified Cas9 ribonucleoproteins. *Genome Res*. 2014;24(6):1012-1019. doi:10.1101/gr.171322.113
2. Liang X, Potter J, Kumar S, et al. Rapid and highly efficient mammalian cell engineering via Cas9 protein transfection. *J Biotechnol*. 2015;208:44-53. doi:10.1016/j.jbiotec.2015.04.024
3. Clement K, Rees H, Canver MC, et al. CRISPResso2 provides accurate and rapid genome editing sequence analysis. *Nat Biotechnol*. 2019;37(3):224-226. doi:10.1038/s41587-019-0032-3
4. Tsai SQ, Zheng Z, Nguyen NT, et al. GUIDE-Seq enables genome-wide profiling of off-target cleavage by CRISPR-Cas nucleases. *Nat Biotechnol*. 2015;33(2):187. doi:10.1038/NBT.3117
5. Vakulskas CA, Dever DP, Rettig GR, et al. A high-fidelity Cas9 mutant delivered as a ribonucleoprotein complex enables efficient gene editing in human hematopoietic stem and progenitor cells. *Nat Med*. 2018;24(8):1216-1224. doi:10.1038/s41591-018-0137-0
6. Kaepffel C, Beattie SG, Fronza R, et al. A largely random AAV integration profile after LPLD gene therapy. *Nat Med*. 2013;19(7):889-891. doi:10.1038/NM.3230
7. Hanlon KS, Kleinstiver BP, Garcia SP, et al. High levels of AAV vector integration into CRISPR-induced DNA breaks. *Nat Commun*. 2019;10(1). doi:10.1038/s41467-019-12449-2
8. Li C, Lieber A. Adenovirus vectors in hematopoietic stem cell genome editing. *FEBS Lett*. 2019;593(24):3623-3648. doi:10.1002/1873-3468.13668
9. Stephens CJ, Kashentseva E, Everett W, Kaliberova L, Curiel DT. Targeted in vivo knock-in of human alpha-1-antitrypsin cDNA using adenoviral delivery of CRISPR/Cas9. *Gene Ther*. 2018;25(2):139-156. doi:10.1038/S41434-018-0003-1
10. Nokin MJ, Mira A, Patrucco E, et al. RAS-ON inhibition overcomes clinical resistance to KRAS G12C-OFF covalent blockade. *Nat Commun*. 2024;15(1):7554. doi:10.1038/s41467-024-51828-2
11. Dy GK, Govindan R, Velcheti V, et al. Long-Term Outcomes and Molecular Correlates of Sotorasib Efficacy in Patients With Pretreated KRAS G12C-Mutated Non-Small-Cell Lung Cancer: 2-Year Analysis of CodeBreak 100. *J Clin Oncol*. 2023;41(18):3311-3317. doi:10.1200/JCO.22.02524
12. Picco G, Petti C, Trusolino L, Bertotti A, Medico E. A diphtheria toxin resistance marker for in vitro and in vivo selection of stably transduced human cells. *Sci Rep*. 2015;5. doi:10.1038/srep14721
13. Øyvind Enger P, Thorsen F, Eystein Lønning P, Bjerkvig R, Hoover F. *Adeno-Associated Viral Vectors Penetrate Human Solid Tumor Tissue In Vivo More Effectively than Adenoviral Vectors*. Vol 13.; 2002.
14. Chen C, Akerstrom V, Baus J, Lan MS, Breslin MB. Comparative analysis of the transduction efficiency of five adeno associated virus serotypes and VSV-G pseudotype lentiviral vector in lung cancer cells. *Virol J*. 2013;10. doi:10.1186/1743-422X-10-86
15. Negro M V, Araujo HA, Lamberti G, et al. Comutations and KRASG12C Inhibitor Efficacy in Advanced NSCLC. *Cancer Discov*. 2023;13(7):1556-1571. doi:10.1158/2159-8290.CD-22-1420

16. Sentmanat MF, Peters ST, Florian CP, Connelly JP, Pruett-Miller SM. A Survey of Validation Strategies for CRISPR-Cas9 Editing. *Sci Rep.* 2018;8(1). doi:10.1038/s41598-018-19441-8
17. Guschin DY, Waite AJ, Katibah GE, Miller JC, Holmes MC, Rebar EJ. A Rapid and General Assay for Monitoring Endogenous Gene Modification. In: ; 2010:247-256. doi:10.1007/978-1-60761-753-2_15
18. Tao Y, Lamas V, Du W, et al. Treatment of monogenic and digenic dominant genetic hearing loss by CRISPR-Cas9 ribonucleoprotein delivery in vivo. *Nat Commun.* 2023;14(1). doi:10.1038/s41467-023-40476-7
19. Gullinger JP, Thompson DB, Liu DR. Fusion of catalytically inactive Cas9 to FokI nuclease improves the specificity of genome modification. *Nat Biotechnol.* 2014;32(6):577-582. doi:10.1038/nbt.2909
20. Singh A, Greninger P, Rhodes D, et al. A gene expression signature associated with "K-Ras addiction" reveals regulators of EMT and tumor cell survival. *Cancer Cell.* 2009;15(6):489-500. doi:10.1016/J.CCR.2009.03.022
21. Hallin J, Engstrom LD, Hargi L, et al. The KRASG12C inhibitor MRTX849 provides insight toward therapeutic susceptibility of KRAS-mutant cancers in mouse models and patients. *Cancer Discov.* 2020;10(1):54-71. doi:10.1158/2159-8290.CD-19-1167
22. Mohanty A, Nam A, Srivastava S, et al. Acquired resistance to KRAS G12C small-molecule inhibitors via genetic/nongenetic mechanisms in lung cancer. *Sci Adv.* 2023;9(41). doi:10.1126/SCIADV.ADE3816/SUPPL_FILE/SCIADV.ADE3816_DATA_FILES_S1_TO_S3.ZIP
23. Koga T, Suda K, Fujino T, et al. KRAS Secondary Mutations That Confer Acquired Resistance to KRAS G12C Inhibitors, Sotorasib and Adagrasib, and Overcoming Strategies: Insights From In Vitro Experiments. *Journal of Thoracic Oncology.* 2021;16(8):1321-1332. doi:10.1016/J.JTHO.2021.04.015